# Accelerating Fair Federated Learning:
# Adaptive Federated Adam

## Abstract

Federated learning is a distributed and privacy-preserving approach to train a statistical model collaboratively from decentralized data of different parties. However, when datasets of participants are not independent and identically distributed, models trained by naive federated algorithms may be biased towards certain participants, and model performance across participants is non-uniform. This is known as the fairness problem in federated learning. In this paper, we formulate fairness-controlled federated learning as a dynamical multi-objective optimization problem to ensure fair performance across all participants. To solve the problem efficiently, we study the convergence and bias of `Adam` as the server optimizer in federated learning, and propose Adaptive Federated Adam (`AdaFedAdam`) to accelerate fair federated learning with alleviated bias. We validated the effectiveness, Pareto optimality and robustness of `AdaFedAdam` with numerical experiments and show that `AdaFedAdam` outperforms existing algorithms, providing better convergence and fairness properties of the federated scheme.

## 1 Introduction

Federated Learning (FL), first proposed by McMahan et al. (2017), is an emerging collaborative learning technique enabling multiple parties to train a joint machine learning model with input privacy being preserved. By iteratively aggregating local model updates done by participating clients using their local, private data, a joint global model is obtained. The promise of federated learning is to achieve superior performance compared to models trained in isolation by each participant. Compared with traditional distributed machine learning, FL works with larger local updates and seeks to minimize communication cost while keeping the data of participants local and private. With increasing concerns about data security and privacy protection, federated learning has attracted much research interest (Li et al., 2020b; Kairouz et al., 2021) and has been proven to work effectively in various application domains (Li et al., 2020a; Xu et al., 2021).

When datasets at the client sites are not independent and identically distributed (IID), the standard algorithm for federated learning, `FedAvg`, can struggle to achieve good model performance, with an increase of communication rounds needed for convergence (Li et al., 2019b; Zhu et al., 2021). Moreover, the global model trained with heterogeneous data can be biased towards some of the participants, while performing poorly for others (Mohri et al., 2019). This is known as fairness problem in federated learning. The term *fairness* encompasses various interpretations such as group fairness and performance fairness in the field of machine learning(Mehrabi et al., 2021). In this paper, we specifically employ the term *fairness* to describe disparities in model performance observed among participants in a federated learning process. There are ways to improve fairness in federated learning, at the cost of model convergence (Mohri et al., 2019; Li et al., 2019a; 2021; Hu et al., 2022). This study aims to contribute to enabling fair federated learning without negatively impacting the convergence rate.

Acceleration techniques for federated learning aim at reducing the communication cost and improving convergence. For instance, momentum-based and adaptive optimization methods such as `AdaGrad`, `Adam`, Momentum `SGD` have been applied to accelerate the training process (Wang et al., 2019; Karimireddy et al., 2020; Reddi et al., 2020). However, default hyperparameters of adaptive optimizers tuned for centralized training

do not tend to perform well in federated settings (Reddi et al., 2020). Furthermore, optimal hyperparameters are not generalizable for federated learning, and hyperparameter optimization with e.g. grid search is needed for each specific federated task, which is infeasible due to the expensive (sometimes unbounded) nature for federated learning. Further research is needed to understand how to adapt optimizers for federated learning with minimal hyperparameter selection.

In this study, to accelerate the training of fair federated learning, we formulate fairness-aware federated learning as a *dynamical multi-objective optimization problem* (DMOO) problem. By analyzing the convergence and bias of federated `Adam`, we propose Adaptive Federated Adam (`AdaFedAdam`) to solve the formulated DMOO problem efficiently. With experiments on standard benchmark datasets, we illustrate that `AdaFedAdam` alleviates model unfairness and accelerates federated training. In additional, `AdaFedAdam` is proved to be robust against different levels of data and resource heterogeneity, which suggests that its performance on fair federated learning can be expected in real-life use cases.

The remainder of the paper is structured as follows. Section 2 summarizes related work including different acceleration techniques for federated training and the fairness problem in federated learning. Then fair federated learning problem is formulated in Section 3 and Federated `Adam` is analyzed in Section 4. Section 5 introduces the design of `AdaFedAdam` and shows the convergence guarantee for `AdaFedAdam`. Experimental setups and results are presented in Section 6. Finally, Section 7 concludes the paper and suggests future research directions.

## 2 Related Work

In this section, we review recent techniques to accelerate federated training as well as studies of model fairness in FL.

### 2.1 Acceleration Techniques for Federated Learning

Adaptive methods and momentum-based methods accelerate centralized training of neural networks over vanilla `SGD` (Kingma & Ba, 2014; Zeiler, 2012). In the context of federated learning, Hsu et al. (2019) and Wang et al. (2019) introduced first-order momentum to update the global model by treating local updates as pseudo-gradients, showing the effectiveness of adaptive methods for federated learning. Further, Reddi et al. (2020) demonstrated a two-level optimization framework `FedOpt` for federated optimization. On the local level, clients optimize the local objective functions while local updates are aggregated as "pseudo-gradients" to update the global model on the server level. By applying adaptive optimizers (e.g. `Adam`) as the server optimizer, we obtain adaptive federated optimizers (e.g. `FedAdam`). It has been empirically validated that adaptive federated optimizers are able to accelerate training with careful fine-tuning (Reddi et al., 2020).

Fine-tuning for server optimizers is challenging for the following reasons:

- Due to the inherent differences of federated and centralized training, default hyperparameters of optimizers which work well on centralized training does not necessarily have satisfactory performance in federated training.

- For adaptive optimizers, grid search needs to be performed to get multiple hyperparameters optimized (Reddi et al. (2020)), which is prohibitively expensive considering the orchestration cost for the entire federation.

- The optimal hyperparameters for server-side optimizers are not generalizable across different federated tasks, and fine-tuning must be done for each individual task.

It would greatly ease the use of server-side optimizers if the selection of hyperparameters were automatic. The proposed method `AdaFedAdam` minimizes the efforts of fine-tuning by adapting default hyperparameters of `Adam` in centralized settings to federated training.

## 2.2 Model Fairness

The concept of model unfairness describes the differences of model performance across participants (Mohri et al., 2019). In a federated training process, model performances of clients may be not uniform when data across participants are heterogeneous, and the global model can be biased towards some participants. To reduce the unfairness, Mohri et al. (2019) proposed the algorithm `Agnostic Federated Learning`, a minimax optimization approach that only optimizes the single device with the worst performance. Inspired by fair resource allocation, Li et al. (2019a) formulated fair federated learning as a fairness-controlled optimization problem with $\alpha$-fairness function (Mo & Walrand, 2000). An algorithm `q-FedAvg` was proposed to solve the optimization problem, which dynamically adjust step sizes of local `SGD` by iteratively estimating Lipschitz constants. More recently, Hu et al. (2022) interpreted federated learning as a multi-objective optimization problem, and adapted Multi-Gradient Descent Algorithm to federated settings as `FedMGDA+` to reduce the unfairness. Alternatively, Li et al. (2021) proposed `Ditto` to improve the performance fairness by personalizing global models on client sites.

Unlike previous work that are based on `FedAvg` with improved fairness at a cost of model convergence, the here proposed approach formulates fair federated learning as a dynamic multi-objective function and proposes `AdaFedAdam` to solve the formulated problem. Compared with other `FedAvg`-based algorithms for fairness control, `AdaFedAdam` offer equivalent fairness guarantee with improved convergence properties.

## 3 Preliminaries & Problem Formulation

### 3.1 Standard Federated Learning

Considering the distributed optimization problem to minimize the global loss function $F(\mathbf{x})$ across $K$ clients as follows:

$$\min_{\mathbf{x}}[F(\mathbf{x}) := \sum_{k=1}^{K} p_k F_k(\mathbf{x})] \tag{1}$$

where $\mathbf{x}$ denotes the parameter set of function $F$, $F_k(\mathbf{x})$ is the local objective function of client $k$ w.r.t local dataset $D_k$, and $p_k := \frac{|D_k|}{\sum |D|}$ denotes the relative sample size of $D_k$ with number of samples $|D_k|$ in $D_k$. The data distribution on client $k$ is denoted by $\mathcal{D}_k$.

### 3.2 Fair Federated Learning

When $D_k$ across clients are non-IID, the standard formulation of federated learning can suffer from significant fairness problem (Mohri et al., 2019) in addition to a potential loss of convergence. To improve the model fairness, federated learning can be formulated with $\alpha$-fairness function as $\alpha$-fair Federated Learning (also known as $q$-fair Federated Learning in Li et al. (2019a)) as follows:

$$\min_{\mathbf{x}}[F(\mathbf{x}) := \sum_{k=1}^{K} \frac{p_k}{\alpha + 1} F_k^{\alpha+1}(\mathbf{x})] \tag{2}$$

where notations is as in equation 1 with additional $\alpha \geq 0$.

However, it is challenging to solve the problem with distributed first-order optimization. With only access to gradients of local objective functions $\nabla^t F_k(\mathbf{x})$, the gradient of $F(\mathbf{x})$ at $\mathbf{x}^t$ and the update rule of distributed `SGD` are given as follows:

$$\nabla F(\mathbf{x}^t) = \sum_{k=1}^{K} p_k F_k^{\alpha}(\mathbf{x}^t) \nabla F_k(\mathbf{x}^t) \tag{3}$$

$$\mathbf{x}^{t+1} := \mathbf{x}^t - \eta \nabla F(\mathbf{x}^t) \tag{4}$$

The gradient $\nabla F(\mathbf{x}^t)$ has decreasing scales due to the factor $F^\alpha(\mathbf{x}^t)$. With the number of iterations $t$ increases, decreasing $F^\alpha(\mathbf{x}^t)$ scales $\nabla F(\mathbf{x}^t)$ down drastically. With a fixed learning rate $\eta$, the update of SGD $-\eta\nabla F(\mathbf{x}^t)$ scales down correspondingly and thus, the convergence deteriorates. To improve the convergence, Li et al. (2019a) proposes q-FedAvg to adjust learning rates adaptively to alleviate the problem. However, the convergence of q-FedAvg is still slow since the scaling challenge is introduced by the problem formulation intrinsically. To achieve a better convergence for first-order federated optimization methods with fairness control, reformulating the problem is required.

### 3.3 Problem Formulation

In the field of multi-task learning, neural networks are designed to achieve multiple tasks at the same time by summing multiple component objective functions up as a joint loss function. In a similar spirit to fair federated learning, training multitask deep neural networks also requires keeping similar progress for all component objectives. Inspired by Chen et al. (2018), we formulate fair federated learning as a dynamic multi-objective optimization problem in the following form:

$$\min_{\mathbf{x}}[F(\mathbf{x}, t) := \frac{\sum_{k=1}^K p_k I_k^\alpha(t) F_k(\mathbf{x})}{\sum_{k=1}^K p_k I_k^\alpha(t)}] \tag{5}$$

Here, the notations are the same as in Equation equation 1. Additionally, *inverse training rate* is defined as $I_k(t) := F_k(\mathbf{x}^t)/F_k(\mathbf{x}^0)$ for client $k$ at round $t$, to quantify its training progress. $\alpha \geq 0$ is a hyperparameter to adjust the model fairness similar to $\alpha$ in $\alpha$-fairness function. The problem reduces to the federated optimization without fairness control if setting $\alpha = 0$, and it restores the minimax approach for multi-objective optimization (Mohri et al., 2019) if setting $\alpha$ a sufficiently large value.

Compared with $\alpha$-Fair Federated Learning, the proposed formulation has equivalent fairness guarantee without the problem of decreasing scales of gradients. With a global model $\mathbf{x}^0$ initialized with random weights, we assume that $F_i(\mathbf{x}^0) \simeq F_j(\mathbf{x}^0)$ for $\forall i, j \in [K]$. Then the gradient of $F(\mathbf{x}, t)$ at $\mathbf{x}^t$ is given by:

$$\nabla F(\mathbf{x}^t, t) \simeq \frac{\sum_{k=1}^K p_k F_k^\alpha(\mathbf{x}^t)\nabla F_k(\mathbf{x}^t)}{\sum_{k=1}^K p_k F_k^\alpha(\mathbf{x}^t)} \tag{6}$$

$$\propto \sum_{k=1}^K p_k F_k^\alpha(\mathbf{x}^t)\nabla F_k(\mathbf{x}^t) \tag{7}$$

The gradient $F(\mathbf{x}^t, t)$ of the DMOP formulation is proportional to the gradient of the $\alpha$-Fair Federated Learning equation 2. Thus, with first-order optimization methods, the solution of the DMOP formulation is also the solution of the $\alpha$-fairness function, which has been proved to enjoy $(p, \alpha)$-*Proportional Fairness* (Mo & Walrand, 2000). Moreover, the DMOP formulation of Fair Federated Learning does not have the problem of decreasing gradient scales in the $\alpha$-fairness function, so that distributed first-order optimization methods can be applied to solve the problem more efficiently.

## 4 Analysis of FedAdam

In this section, we analyze the performance of Adam as the server optimizer in federated learning. We first study the effect of using accumulated updates as pseudo-gradients for Adam in centralized training. The bias introduced by averaging accumulated local updates without normalization in FedAdam is then discussed.

### 4.1 From Adam to FedAdam

As the de facto optimizer for centralized deep learning, Adam provides stable performance with little need of fine-tuning. Adam provides adaptive stepsize selection based on the initial stepsize $\eta$ for each individual

coordinate of model weights. The adaptivity of stepsizes can be understood as continuously establishing *trust regions* based on estimations of the first- and second-order momentum(Kingma & Ba, 2014), which are updated by exponential moving averages of gradient estimations and their squares with hyperparameters $\beta_1$ and $\beta_2$ respectively in each step.

The choice of hyperparameters in `Adam` can be explained by the certainty of directions for model updates. In centralized `Adam`, directions for updates are from gradient estimations $\nabla_{\zeta \sim \mathcal{D}} F(\mathbf{x})$ obtained from a batch of data $\zeta$. Generally the size of the batch $|\zeta|$ is relatively small, and the variance of $\nabla_{\zeta \sim \mathcal{D}} F(\mathbf{x})$ is large, indicating low certainty of update directions. Thus, large $\beta_1$ and $\beta_2$ (0.9 and 0.999 by default) are set to assign less weight for each gradient estimation when updating first- and second-order momentum. Low certainty of update directions also only allows small *trust regions* to be constructed from small initial stepsize $\eta$ (default value: 0.001).

In federated learning, `FedAdam` is obtained if we apply `Adam` as the server optimizer. The size-weighted average of clients' local updates at round $t$, $\Delta_t$, acts as the pseudo-gradient. Although empirical results have shown that `FedAdam` outperforms the standard `FedAvg` with careful fine-tuning in terms of average error (Reddi et al., 2020), several problems exist in `FedAdam`. In the following subsections, we analyze the problem of convergence loss of `FedAdam` and bias of the pseudo-gradients used for `FedAdam`.

## 4.2 Adam with Accumulated Updates

When data between clients are statistically homogeneous ($\forall k, i \in [K], \mathcal{D}_k = \mathcal{D}_i$), it has been proven in prior work(Khaled et al., 2020; Stich, 2018) `FedAvg` converges to the same optima as mini-batch SGD, from which within a squared distance of $\mathcal{O}(N-1)$, where $N$ denotes the number of local steps. Thus, the average of local updates is an unbiased estimator of accumulated updates of multiple centralized `SGD` steps. By applying `Adam` as the server optimizer in IID cases, `FedAdam` shrinks as `Adam` with gradient estimation given by accumulated updates of $N$ `SGD` steps ($N$-`AccAdam`). The Pseudo-codes of $N$-`AccAdam` is given in the Appendix B. We prove that even in centralized settings, $N$-`AccAdam` has less convergence guarantee than standard `Adam` with same hyperparameters.

**Theorem 1 (Convergence of $N$-`AccAdam`)** *Assume the L-smooth convex loss function $F(\mathbf{x})$ has bounded gradients $\|\nabla\|_\infty \leq G$ for all $\mathbf{x} \in \mathbb{R}^d$. Hyperparameters $\epsilon$, $\beta_2$ and $\eta$ in $N$-`AccAdam` are chosen with the following conditions: $\eta \leq \epsilon/2L$ and $1 - \beta_2 \leq \epsilon^2/16G^2$. The accumulated update of $N$ SGD updates at step $t$ $\Delta \mathbf{x}^t := -\sum_{n=1}^{N} \Delta_n \mathbf{x}^t$ is applied to Adam, where $\Delta_n \mathbf{x}^t$ denotes the local update after $n$ SGD steps with a fixed learning rate on model $\mathbf{x}^t$. `SGD` exhibits linear convergence to the neighbourhood of the optima with constants $(A, c)$. In the worst case, the algorithm has no convergence guarantee. In the best cases where $R_t = N$ for all $t \in [T]$, the converge guarantee is given by:*

$$\frac{1}{T} \sum_{t=1}^{T} \|\nabla F(\mathbf{x}^t)\|^2 \leq \frac{F(\mathbf{x}^1) - F(\mathbf{x}^*)(\sqrt{\beta_2}G + \epsilon)}{\underbrace{\left(\dfrac{Nc}{1 - (1-c)^N} - \dfrac{1}{2}\right)}_{S} \eta T} \tag{8}$$

*where $R_t := \min |\frac{\Delta_{t,i}}{\nabla_{t,i}}|$ for $i \in [d]$*

The proof for Theorem 1 is deferred to Appendix A.1. In the best case where $R_t = N$ (which is almost not feasible), $N$-`AccAdam` gains $S$ a speedup compared with `Adam`. However, the computation cost of $N$-`AccAdam` is linear to $N$ but the speedup $S$ is sublinear with respect to $N$. Thus, with a fixed computation budget, the convergence rate of $N$-`AccAdam` is slower than `Adam` with the same hyperparameters. Compared with gradient estimation by a small batch of data, accumulated updates of multiple `SGD` steps have larger certainty about directions of updates for the global model. To improve the convergence of $N$-`AccAdam`, it is possible to construct larger *trust regions* with larger stepsize $\eta$ and smaller $\beta$ values with accumulated updates.

### 4.3 Bias of Pseudo-gradients for `FedAdam`

In federated settings, when data among clients are heterogeneous, the average of local updates weighted by sizes of client datasets introduces bias toward a portion of clients. Moreover, the biased pseudo-gradients lead to even lower convergence and increase the unfairness of `FedAdam`.

Wang et al. (2020) has proved that there exists objective inconsistency between the stationary point and the global objective function, and biases are caused by different local `SGD` steps taken by clients. They propose `FedNova` to reduce the inconsistency by normalizing local updates with the number of local steps. The convergence analysis of `FedNova` assumes that all local objective functions have the same $L$-smoothness, which is also identical to the smoothness constant of the global objective function. However, in federated learning with highly heterogeneous datasets, smoothness constants $L_k$ of local objective functions $F_k(\mathbf{x})$ are very different across clients and from $L_g$ of the global objective function. Although the assumption and proof still holds if taking $L_g := \max(L_k)$ for all $k \in [K]$, we argue that the inconsistency still exists in `FedNova` if only normalizing local updates with number of steps regardless of differences of $L_k$-smoothness constant of local objectives.

In one communication round, with the same numbers of local `SGD` steps and a fixed learning rate $\eta$, it is likely to happen that while objectives with small $L$-constant are still slowly converging, local objectives with large $L$-constants have converged in a few steps and extra steps are ineffective. In such cases, normalizing local updates with number of local `SGD` steps implicitly over-weights updates from objectives with smaller $L$-constants when computing pseudo-gradients. Further improvements are needed in the normalization of local updates to de-bias the pseudo-gradients, accounting for both the different numbers of local steps and the $L$-smoothness constants of local objectives.

## 5 `AdaFedAdam`

To address the drawbacks mentioned above, we propose Adaptive `FedAdam` (`AdaFedAdam`) to make better use of accumulated local updates for fair federated learning (equation 5) with little efforts on fine-tuning.

### 5.1 Algorithm

Figure 1 is an illustration of `AdaFedAdam`. Intuitively, the proposed method `AdaFedAdam` operates as follows: it initially estimates the certainty values of local updates obtained after local training, indicating the confidence of each local update. Subsequently, using the normalized local updates and their associated certainty values, it computes the pseudo-gradient along with its certainty. Adaptive adjustment of hyperparameters is then performed based on the certainty of the pseudo-gradient. Finally, the pseudo-gradient is utilized to update the global model, incorporating the locally trained information of both the update directions and their associated certainties. The pseudo-code of the algorithm is presented as Algorithm 1 and Algorithm 2.

Compared with standard `FedAdam`, three extra steps are taken, as explained in details below:

**Normalization of local updates:** Due to different $L$-smoothness constants of local objectives and local steps across participants, lengths of accumulated updates $\boldsymbol{\Delta}_k$ are not at uniform scales and normalization of local updates is necessary as discussed in Section 4. Natural scales for local updates are the $\ell_2$-norms of local gradients $\|\nabla F_k(\mathbf{x}^t)\|_2$ on client $k$. By normalizing $\boldsymbol{\Delta}_k$ to the same $\ell_2$-norm of $\|\nabla F_k(\mathbf{x}^t)\|_2$, a normalized update $\mathbf{U}_k$ and a supportive factor $\eta'_k$ are obtained. Intuitively, $\boldsymbol{\Delta}_k$ can be interpreted as a single update step following a "confident" update direction $-\mathbf{U}_k$ with a large learning rate $\eta'_k$ on the model $\mathbf{x}^t$ provided by client $k$. The certainty of the direction $\mathbf{U}^k$ is defined as $C_k := \log(\eta'_k/\eta_k) + 1$ ($\eta_k$ as the learning rate of the local solver), and the larger value of $C_k$ is, the greater update can be made following $\mathbf{U}_k$.

**Fairness control:** Recall that $I_k$ represents the inverse training rate and $\alpha$ is a predefined hyperparameter for fairness control. Following the formulation of the loss function in fair federated learning in Section 3, the pseudo-gradient $\mathbf{g}^t$ of the global model $\mathbf{x}^t$ is correspondingly the average of the normalized local updates with adaptive weights $I_k^\alpha$. The certainty of $\mathbf{g}^t$ is given by the weighted average of local certainties $C_k$ for all $k \in [K]$.

---

**Algorithm 1** `AdaFedAdam`: Adaptive Federated `Adam`

---

**Require:** initial model $\mathbf{x}^0$, $\eta$, $\beta_1$, $\beta_2$, $\epsilon$

  Initialize $m$ and $v$: $m_0 \leftarrow 0$, $v_0 \leftarrow 0$

  Initialize correction factors for $m$ and $v$: $c_{0,m} \leftarrow 1$, $c_{0,v} \leftarrow 1$

  **for** round $t$ in $\{0, 1, ...T-1\}$ **do**

    $\mathbf{g}^t, C^t = \textbf{GetPseudoGradient}(\mathbf{x}^t)$             $\triangleright$ Compute pseudo-gradient and its certainty

    $\beta_{t,1} \leftarrow \beta_1^{C^t}$, $\beta_{t,2} \leftarrow \beta_2^{C^t}$                              $\triangleright$ Adapt $\beta_1$ and $\beta_2$

    $\eta_t \leftarrow C^t \eta$                                          $\triangleright$ Adapt step size $\eta$

    $c_{t+1,m} \leftarrow c_{t,m}\beta_{t,1}$, $c_{t+1,v} \leftarrow c_{t,v}\beta_{t,2}$          $\triangleright$ Update correction factors

    $m_{t+1} \leftarrow (1-\beta_{t,1})\mathbf{g}^t + \beta_{t,1}m_t$

    $v_{t+1} \leftarrow (1-\beta_{t,2})\mathbf{g}^t \odot \mathbf{g}^t + \beta_{t,2}v_t$                       $\triangleright$ Update $m$ and $v$

    $\hat{m}_{t+1} \leftarrow m_{t+1}/(1-c_{t+1,m})$

    $\hat{v}_{t+1} \leftarrow v_{t+1}/(1-c_{t+1,v})$                             $\triangleright$ Correct $m$ and $v$

    $\mathbf{x}^{t+1} \leftarrow \mathbf{x}^t - \eta\hat{m}_{t+1}/(\sqrt{\hat{v}_{t+1}} + \epsilon)$             $\triangleright$ Update model weights

  **end for**

---

**Algorithm 2** Pseudo-gradient calculation

---

**Require:** $\alpha$

  **function** GETPSEUDOGRADIENT($\mathbf{x}$))

    Broadcast $\mathbf{x}$ to all clients

    **for** client $k$ in $\{0, 1, ...K-1\}$ **parallel do**

      Calculate $F_k(\mathbf{x})$ and $\nabla F_k(\mathbf{x})$

      $\mathbf{x}_k = \textbf{LocalSolver}(\mathbf{x}, \eta_k)$                   $\triangleright$ Local training on client datasets

      $\boldsymbol{\Delta}_k \leftarrow \mathbf{x}_k - \mathbf{x}$

      $\eta'_k \leftarrow \frac{\|\boldsymbol{\Delta}_k\|_2}{\|\nabla F_k(\mathbf{x}^t)\|_2}$

      $\mathbf{U}_k \leftarrow -\frac{\boldsymbol{\Delta}_k}{\eta'_k}$                              $\triangleright$ Normalize local updates

      $C_k \leftarrow \log(\eta'_k/\eta_k) + 1$                     $\triangleright$ Estimate certainty of local updates

      $I_k = F_k(\mathbf{x})/F_k(\mathbf{x}^0)$

      Report $(\mathbf{U}_k, C_k, I_k)$ to the server

    **end for**

    $\mathbf{g} \leftarrow \frac{\sum p_k I_k^\alpha \mathbf{U}_k}{\sum p_k I_k^\alpha}$, $C \leftarrow \frac{\sum p_k I_k^\alpha C_k}{\sum p_k I_k^\alpha}$           $\triangleright$ Aggregate local updates

    **Return** $\mathbf{g}$, $C$

  **end function**

---

**Adaptive hyperparameters for federated `Adam`:** Hyperparameters of `FedAdam` are adapted as follows to make better use of pseudo-gradients $\mathbf{g}$ from accumulated updates. Here, $C$ represents the certainty of the pseudo-gradient and is determined based on the weighted average of local certainties $C_k$ as discussed earlier.

- $\beta_{t,1} \leftarrow \beta_1^C$, $\beta_{t,2} \leftarrow \beta_2^C$: Adaptive $\beta_{t,1}$ and $\beta_{t,2}$ dynamically control the weight of the current update for the momentum estimation. `AdaFedAdam` assigns more weight to more "certain" pseudo-gradients to update the average, and thus $\beta_1$ and $\beta_2$ are adapted exponentially following the form of exponentially weighted moving average.

- $\eta_t \leftarrow C\eta$: The base stepsize $\eta_t$ is adjusted based on the certainty of the pseudo-gradient $C$ as well. Greater certainty enables larger $\eta_t$ to construct larger *trust regions* and vice versa.

Theoretically, `AdaFedAdam` ensures the following features:

- **Fairness Guarantee**: The fairness of the model has been incorporated into the objective function, optimizing both the error and the fairness with theoretical $(p, \alpha)$-*Proportional Fairness* (Mo & Walrand, 2000) guarantee. Also, the algorithm can be adapted to different fairness levels by adjusting $\alpha$ in the problem formulation.

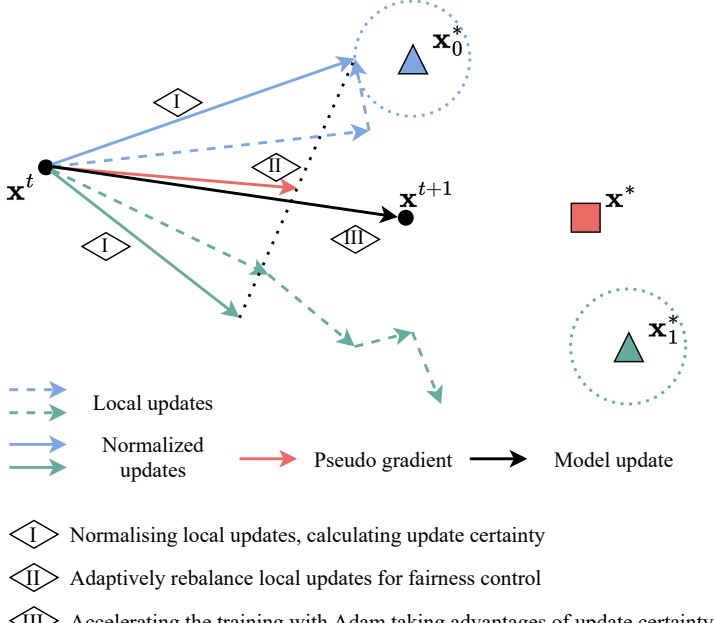

Figure 1: Illustration of `AdaFedAdam`: $\mathbf{x}^t$, $\mathbf{x}_k^*$ and $\mathbf{x}^*$ denote the global model at round $t$, the optima of local objective functions of client $k$ and the optima of the global objective function, respectively.

- **Fine-tuning Free:** The adaptivity of `AdaFedAdam` derives from dynamic adjustment of hyperparameters for `Adam`. All initial hyperparameters of `AdaFedAdam` can be chosen as the default values in the standard `Adam` for the centralized setting, and they are adaptively adjusted during the federated training process.

- **Allowance for Resource Heterogeneity:** Thanks to the normalization of local updates, `AdaFedAdam` allows arbitrary numbers of local steps, which could be caused by resource limitation of clients (also known as resource heterogeneity).

- **Compatibility with Arbitrary Local Solvers:** The normalization of local updates only relies on the $\ell_2$-norm of the local gradient estimation. Thus, any first-order optimizers are compatible with `AdaFedAdam`.

These features of `AdaFedAdam` are empirically validated and discussed in Section 6.

## 5.2 Convergence analysis for `AdaFedAdam`

The convergence guarantee of `AdaFedAdam` for convex functions is proved as follows.

**Theorem 2 (Convergence of `AdaFedAdam`)** *Assume the $L$-smooth convex loss function $F(\mathbf{x})$ has bounded gradients $\|\nabla\|_\infty \leq G$ for all $\mathbf{x} \in \mathbb{R}^d$, and hyperparameters $\epsilon$, $\beta_{2,0}$ and $\eta_0$ are chosen according to the following conditions: $\eta_0 \leq \epsilon/2L$ and $1 - \beta_{2,0} \leq \epsilon^2/16G^2$. The pseudo-gradient $\mathbf{g}_t$ at step $t$ is given by Algorithm 1 with its certainty $C^t$. The convergence guarantee of `AdaFedAdam` is given by:*

$$\frac{1}{T}\sum_{t=1}^{T}\|\nabla F(\mathbf{x}^t)\|^2 \leq \frac{2(F(\mathbf{x}^1) - F(\mathbf{x}^*))(\sqrt{\beta_{2,0}}G + \epsilon)}{RC\eta_0 T} \tag{9}$$

*where $R := \min_t(\min_i |\frac{\mathbf{g}_{t,i}}{\nabla_{t,i}}|)$ for all $i \in [d], t \in [T]$ and $C := \min C^t$ for $t \in [T]$.*

The proof of Theorem 2 is deferred to Appendix A.2. By normalizing local updates to the same $\ell_2$-norm of local gradients, the convergence of `AdaFedAdam` can be guaranteed. When the client optimizers are fixed

as `SGD` and perform one step locally, the federated training is identical to minibatch `Adam`. Theorem 2 then provides the same convergence guarantee of `Adam` (Reddi et al., 2019). It is important to note that Theorem 2 focuses on the convergence guarantee, rather than providing a tight bound for the convergence rate. Better empirical performance of `AdaFedAdam` can be anticipated.

## 6 EXPERIMENTAL RESULTS

**Experimental Setups** To validates the effectiveness and robustness of `AdaFedAdam`, we conducted experiments with four federated setups: 1). Femnist setup: A multi-layer perceptron (MLP) network (Pal & Mitra, 1992) for image classification on Federated EMNIST dataset (Deng, 2012), proposed by Caldas et al. (2018) as a benchmark task for federated learning; 2). Cifar10 setup: VGG11 (Simonyan & Zisserman, 2014) for image classification on Cifar10 dataset (Krizhevsky et al., 2009) partitioned by Dirichlet distribution **Dir**(0.1) for 16 clients; 3). Sent140 setup: A stacked-LSTM model (Gers et al., 2000) for sentiment analysis on the Text Dataset of Tweets (Go et al., 2009); 4). Synthetic setup: A linear regression classifier for multi-class classification on a synthetic dataset (Synthetic), proposed by Caldas et al. (2018) as a challenging task for benchmarking federated algorithms. Details of the model architectures, experimental settings and fine-tuning methods are available in Appendix C.1. All code, data, and experiments are publicly available at GitHub (the link to be provided in the final manuscript).

**Convergence & Fairness** We benchmark `AdaFedAdam` against `FedAvg`, `FedAdam`, `FedNova`, `FedProx` with optimal $\mu$ and `q-FedAvg` with $q = 1$. To evaluate the fairness of the models, we measured the standard deviation (STD) of local accuracy on the clients and the average accuracy of the worst 30% clients. The training curves are presented in Figure 2 with numerical details in Appendix C.2. Figure 2 shows that `AdaFedAdam` consistently converges faster than other algorithms, with better worst 30% client performance for all setups. Probability density of distributions of local accuracy indicates that federated models trained with `AdaFedAdam` provide local accuracy with the least standard deviation for the participants. In contrast, it is observed that other federated algorithms do not provide consistent performance in different setups with fine-tuning. To summarize, even without fine-tuning, `AdaFedAdam` is able to train fair federated models among participants with better convergence.

**Choice of $\alpha$** Hyperparameter $\alpha$ is to control the level of desired model fairness. By increasing $\alpha$, models become more fair between clients at a cost of convergence. Thus, for each federated learning process, there exists a Pareto Front Ngatchou et al. (2005) for the trade-off. Taken the Synthetic setup as an example, the average and relative standard deviation (RSD, definted as $\frac{\sigma}{\mu}$) of local validation error during the training process and the formed Pareto Front is shown as Figure 3. It is observed that with increase of $\alpha$ from 1 to 4, the RSD of the local error decreases significantly with a slight decrease of the convergence. With $\alpha > 4$, the RSD of the local error does not reduce significantly but the convergence continues to decrease. By plotting the average and RDS of local error of models trained with `AdaFedAdam` for different $\alpha$ together with other federated algorithms, it can be observed that `FedAvg`, `FedAdam`, `FedNova`, `FedProx` and `q-FedAvg` (with $q \in \{1, 2, 4\}$) are sub-optimal in the blue area in Figure 3. By default, $1 \leq \alpha \leq 4$ is enough to provide proper model fairness without sacrificing much convergence speed.

**Robustness** Experiments to validate the robustness of `AdaFedAdam` against resource heterogeneity and different levels of data heterogeneity are conducted with the Cifar10 setup.

Robustness against resource heterogeneity is important for algorithms to be applied in real life. Due to the heterogeneity of clients' computing resources, the server cannot expect all participants perform requested number of local steps / epochs in each global round and thus, clients may perform arbitrary numbers of local steps on the global model in each communication round. To simulate settings of resource heterogeneity, time-varying numbers of local epochs are randomly sampled from a uniform distribution $\mathcal{U}(1, 3)$ in each communication round for each participant. The results are shonw in Table 1. With resource heterogeneity, `AdaFedAdam` outperform other federated algorithms with higher average accuracy and more fairness.

Robustness against different non-IID levels ensures the performance of an algorithm in various application cases. To simulate different non-IID levels, the Cifar10 dataset is partitioned by the Dirichlet distribution

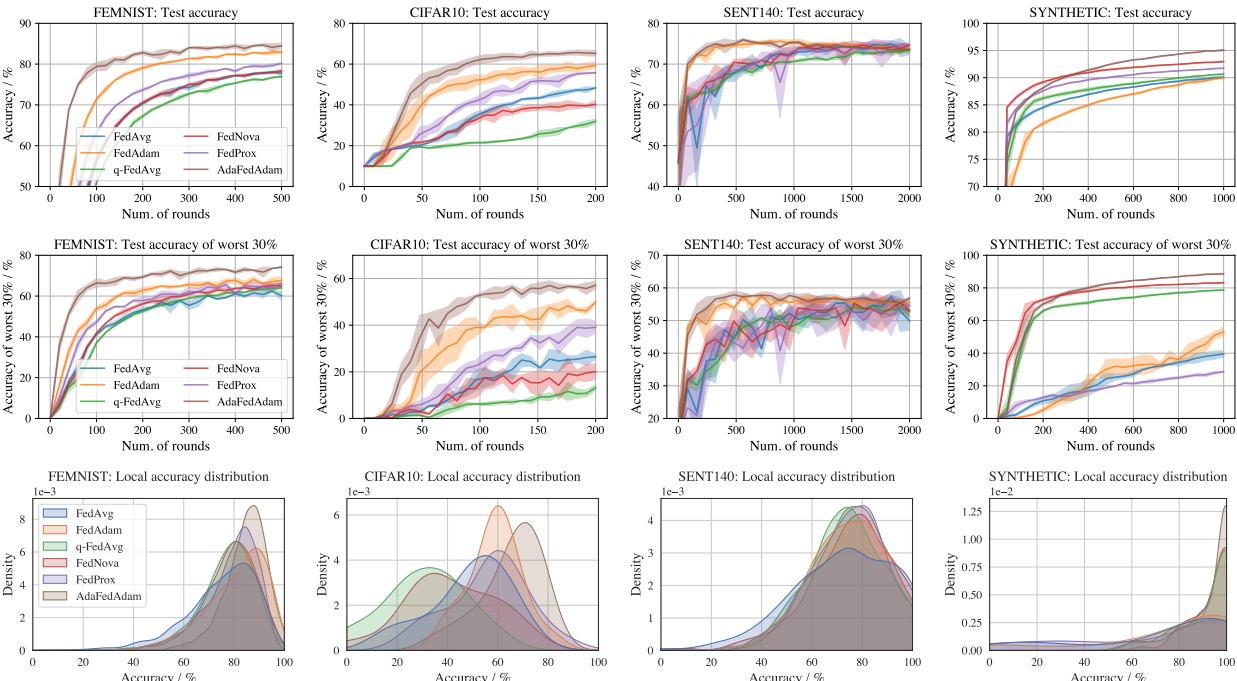

Figure 2: Metrics of local test accuracy during the training process: `FedAvg`, `FedAdam`, `FedNova`, `FedProx`, `q-FedAvg` and `AdaFedAdam` on different setups. **Top:** Average of local test accuracy of participants. **Middle:** Average of local test accuracy of the worst 30% participants. **Bottom:** Probability density of distributions of local test accuracy. Figures in each row share the same legend in the first figure. For all setups, `AdaFedAdam` consistently outperforms baseline algorithms, with better model convergence and fairness among the participants.

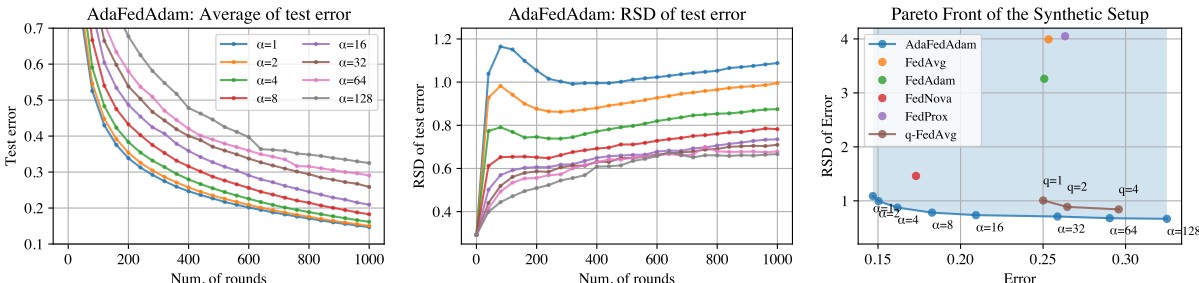

Figure 3: **Left:** Training curves of `AdaFedAdam` with different $\alpha$ on the Synthetic setup. **Middle:** RSD of local error of `AdaFedAdam` with different $\alpha$ on the Synthetic setup. **Right:** Pareto Front of the Synthetic setup formed by `AdaFedAdam` with different $\alpha$. By adjusting values of $\alpha$, the trade-off between the convergence and the fairness can be observed together with the suboptimality of other federated algorithms.

over labels with different concentration parameters $\beta \in [0, 1]$, denoted as $\mathbf{Dir}(\beta)$. A smaller value of $\beta$ indicates larger level of data heterogeneity. The results are shown in Table 2. With different levels of data heterogeneity, `AdaFedAdam` is able to converge, and better performance and fairness are obtained in settings with less data heterogeneity, as expected.

**Compatibility with Momentum-based Local Solvers**   We also show that `AdaFedAdam` is compatible with momentum-based local optimizers in Table 3, which can further improve the model performance. Adaptive optimizers as client solvers (e.g. `Adam`) do not guarantee better performance over vanilla `SGD` without

Table 1: Experimental result of federated algorithms against resource heterogeneity on the Cifar10 setup. Time-varying numbers of local epochs are sampled from a uniform distribution $\mathcal{U}(1, 3)$ in each communication round. Test accuracy of models is reported.

| Algorithm | Avg.(%) | STD.(%) | Worst 30. (%) |
|-----------|---------|---------|---------------|
| FedAvg | 49.31 ±1.37 | 12.28 ±1.81 | 33.55 ±2.29 |
| FedAdam | 62.44 ±2.06 | 16.11 ±2.53 | 42.25 ±2.11 |
| q-FedAvg | 30.96 ±1.67 | 15.69 ±0.95 | 11.58 ±1.16 |
| FedNova | 45.77 ±3.27 | 20.1 ±2.15 | 25.24 ±2.71 |
| FedProx | 58.25 ±1.1 | 12.76 ±1.23 | 39.95 ±2.78 |
| AdaFedAdam | 67.59 ±1.61 | 8.5 ±2.2 | 58.01 ±2.38 |

Table 2: Experimental results of `AdaFedAdam` in different levels of non-iid settings on the Cifar10 setup. Test accuracy of models is reported.

| Distribution | Avg.(%) | STD.(%) | Worst 30. (%) |
|--------------|---------|---------|---------------|
| **Dir**(0.05) | 62.81 ±1.02 | 8.18 ±1.33 | 46.01 ±2.23 |
| **Dir**(0.1) | 66.16 ±1.13 | 6.58 ±0.77 | 55.26 ±2.83 |
| **Dir**(0.5) | 71.43 ±0.81 | 5.4 ±0.22 | 64.93 ±1.04 |
| **Dir**(1) | 72.77 ±0.44 | 3.05 ±0.11 | 69.57 ±0.54 |

synchronizing states of local optimizers, as discussed in Yu et al. (2019) and Yuan & Ma (2020). There are reported algorithms to synchronize states of local optimizers and `AdaFedAdam` is orthogonal and compatible with these algorithms. Full experimental results for different local solvers are deferred to Appendix C.2.

Table 3: Experimental results of the `AdaFedAdam` with different local optimizers on the Synthetic setup, including vanilla `SGD`, `SGD` with momentum and `SGD` with Nesterov momentum. Test accuracy of models is reported.

| Local Optimizer | Avg.(%) | STD.(%) | Worst 30. (%) |
|-----------------|---------|---------|---------------|
| Vanilla `SGD` | 94.18 ±0.45 | 8.52 ±0.37 | 87.07 ±2.28 |
| `SGD` w. Momen. | 97.19 ±0.11 | 3.32 ±0.02 | 93.41 ±0.11 |
| `SGD` w. Neste. Momen. | 97.27 ±0.16 | 3.19 ±0.21 | 94.19 ±0.24 |

## 7 Conclusion

In this work, we address the challenge of fair federated learning by formulating it as a dynamic multi-objective optimization problem. To solve the problem efficiently, we presented `AdaFedAdam` to achieve fair model performance among participants. We demonstrate that `AdaFedAdam` reduces biases in `FedAdam` and accelerates the training process of fair federated learning with minimal fine-tuning efforts. Empirically we validated the efficiency and fairness of `AdaFedAdam` and demonstrated its Pareto optimality compared with other alternative federated algorithms. Further, we showed the robustness of `AdaFedAdam` against resource heterogeneity and different levels of data heterogeneity. We have also demonstrated the compatibility of `AdaFedAdam` with other local optimizers. All code, data, and experiments are publicly available at GitHub (the link will be provided in the final manuscript). In future work, we plan to test `AdaFedAdam` in real-world geographically distributed settings, including cross-silo and cross-device scenarios, using production-grade open-source frameworks such as Yang et al. (2019); Ekmefjord et al. (2022).

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

# A    Proof for Theorems

## A.1    Proof for Theorem 1

In this section we provide the proof for Theorem 1.

**Lemma 3 (Path length bound for Stochastic Gradient Descent)** *With same assumptions for function $F(\mathbf{x})$ in Theorem 1, if the SGD iterates with learning rate $\eta$ exhibit approximately linear convergence with constants $(A, c)$ for $N$ steps, then the path length $\mathcal{L}_N := \sum_0^N \|\mathbf{x}^{n+1} - \mathbf{x}^n\|_2$ is bounded as:*

$$\mathcal{L}_N \leq \|\mathbf{x}^0 - \mathbf{x}^*\|_2 \sum_0^N (1-c)^n \eta A L$$

$$\leq \|\mathbf{x}^0 - \mathbf{x}^*\|_2 \frac{1 - (1-c)^N}{c} A L$$

The proof of the lemma can be referred to Gupta et al. (2021).

Here we analyze the convergence with no momentum ($\beta_1 = 0$), and the result can be extended to general cases (Zaheer et al., 2018).

To simplify the notation, we denote $\nabla_{t,i}$ as the $i$th element of the gradient of model $\nabla F(\mathbf{x}_t)$ at round $t$, and $\Delta_{t,i}$ for the $i$th element of $\Delta_t$. The path length of SGD updates for at step $t$ is denoted as $\mathcal{L}_N^t$

Recall that the update rule of $N$-AccAdam is given by

$$\mathbf{x}_{t+1} = \mathbf{x}_t - \eta \frac{\Delta_t}{\sqrt{v_t} + \epsilon}$$

for all $i \in [d]$. Let $R_t := \min |\frac{\Delta_{t,i}}{\nabla_{t,i}}|$ for $i \in [d]$ and $\mathcal{P}_N^t := \frac{\|\Delta_t\|_2}{\|\nabla F(\mathbf{x}^t)\|_2}$. L-smoothness of function $F(\mathbf{x})$ guarantees that

$$F(\mathbf{x}_{t+1}) \leq F(\mathbf{x}_t) + \langle \nabla_t, \mathbf{x}_{t+1} - \mathbf{x}_t \rangle + \frac{L}{2} \|\mathbf{x}_{t+1} - \mathbf{x}_t\|_2^2$$

$$= F(\mathbf{x}_t) - \eta \sum_{i=1}^d (\nabla_{t,i} \cdot \frac{\Delta_{t,i}}{\sqrt{v_{t,i}} + \epsilon}) + \frac{L\eta^2}{2} \sum_{i=1}^d \frac{\Delta_{t,i}^2}{(\sqrt{v_{t,i}} + \epsilon)^2}$$

$$= F(\mathbf{x}_t) - \eta \sum_{i=1}^d (\nabla_{t,i} \cdot (\frac{\Delta_{t,i}}{\sqrt{v_{t,i}} + \epsilon} - \frac{R_t \nabla_{t,i}}{\sqrt{\beta_2 v_{t-1,i}} + \epsilon}$$

$$+ \frac{R_t \nabla_{t,i}}{\sqrt{\beta_2 v_{t-1,i}} + \epsilon})) + \frac{L\eta^2}{2} \sum_{i=1}^d \frac{\Delta_{t,i}^2}{(\sqrt{v_{t,i}} + \epsilon)^2}$$

$$\leq F(\mathbf{x}_t) - R_t \eta \sum_{i=1}^d \frac{\nabla_{t,i}^2}{\sqrt{\beta_2 v_{t-1,i}} + \epsilon}$$

$$+ \eta \sum_{i=1}^d \nabla_{t,i} \underbrace{|\frac{\Delta_{t,i}}{\sqrt{v_{t,i}} + \epsilon} - \frac{R_t \nabla_{t,i}}{\sqrt{\beta_2 v_{t-1,i}} + \epsilon}|}_{T}$$

$$+ \frac{L\eta^2}{2} \sum_{i=1}^d \frac{\Delta_{t,i}^2}{(\sqrt{v_{t,i}} + \epsilon)^2}$$

$T$ is bounded by

$$
\begin{aligned}
T &= \left| \frac{\Delta_{t,i}}{\sqrt{v_{t,i}} + \epsilon} - \frac{R_t \nabla_{t,i}}{\sqrt{\beta_2 v_{t-1,i}} + \epsilon} \right| \\
&\leq \left| \frac{\Delta_{t,i}}{\sqrt{v_{t,i}} + \epsilon} - \frac{\Delta_{t,i}}{\sqrt{\beta_2 v_{t-1,i}} + \epsilon} \right| \\
&\leq |\Delta_{t,i}| \cdot \left| \frac{1}{\sqrt{v_{t,i}} + \epsilon} - \frac{1}{\sqrt{\beta_2 v_{t-1,i}} + \epsilon} \right| \\
&= \frac{|\Delta_{t,i}|}{(\sqrt{v_{t,i}} + \epsilon)(\sqrt{\beta_2 v_{t-1,i}} + \epsilon)} \cdot \frac{(1-\beta_2)\Delta_{t,i}^2}{\sqrt{v_{t,i}} + \sqrt{\beta_2 v_{t-1,i}}} \\
&\leq \frac{1}{(\sqrt{v_{t,i}} + \epsilon)(\sqrt{\beta_2 v_{t-1,i}} + \epsilon)} \cdot \sqrt{1-\beta_2}\Delta_{t,i}^2 \\
&\leq \frac{\sqrt{1-\beta_2}\Delta_{t,i}^2}{(\sqrt{\beta_2 v_{t-1,i}} + \epsilon)\epsilon}
\end{aligned}
$$

With the bound above and $\|\nabla F(\mathbf{x}_t)\|_\infty \leq G$ for all $i \in [d]$, we have following

$$
\begin{aligned}
F(\mathbf{x}_{t+1}) &\leq F(\mathbf{x}_t) - R_t \eta \sum_{i=1}^d \frac{\nabla_{t,i}^2}{\sqrt{\beta_2 v_{t-1,i}} + \epsilon} \\
&+ \frac{\eta G \sqrt{1-\beta_2}}{\epsilon} \sum_{i=1}^d \frac{\Delta_{t,i}^2}{\sqrt{\beta_2 v_{t-1,i}} + \epsilon} + \frac{L\eta^2}{2\epsilon} \sum_{i=1}^d \frac{\Delta_{t,i}^2}{\sqrt{v_{t,i}} + \epsilon} \\
&\leq F(\mathbf{x}_t) - \eta R_t \sum_{i=1}^d \frac{\nabla_{t,i}^2}{\sqrt{\beta_2 v_{t-1,i}} + \epsilon} \\
&+ \frac{\mathcal{P}_N^t \eta G \sqrt{1-\beta_2}}{\epsilon} \sum_{i=1}^d \frac{\nabla_{t,i}^2}{\sqrt{\beta_2 v_{t-1,i}} + \epsilon} + \frac{\mathcal{P}_N^t L\eta^2}{2\epsilon} \sum_{i=1}^d \frac{\nabla_{t,i}^2}{\sqrt{\beta_2 v_{t-1,i}} + \epsilon}
\end{aligned}
$$

From the parameters $\eta, \epsilon$ and $\beta$ stated in `Adam`, $L\eta/2\epsilon \leq 1/4$ and $G\sqrt{1-\beta_2}/\epsilon \leq 1/4$ hold. Using the inequality conditions and let $V_t := \frac{\|\Delta_t\|_2}{\|\nabla F(\mathbf{x}_t)\|_2}$, we have

$$
\begin{aligned}
F(\mathbf{x}_{t+1}) &\leq F(\mathbf{x}_t) - (R_t - \frac{\mathcal{P}_N^t}{2})\eta \sum_{i=1}^d \frac{\nabla_{t,i}^2}{\sqrt{\beta_2 v_{t-1,i}} + \epsilon} \\
&\leq F(\mathbf{x}_t) - (\frac{R_t}{\mathcal{P}_N^t} - \frac{1}{2})\frac{\eta}{\sqrt{\beta_2}G + \epsilon}\|\nabla F(\mathbf{x}_t)\|^2
\end{aligned}
$$

Using a telescope sum and rearranging the inequality, we have

$$
\frac{\eta}{\sqrt{\beta_2}G + \epsilon} \sum_{t=1}^T (\frac{R_t}{\mathcal{P}_N^t} - \frac{1}{2})\|\nabla F(\mathbf{x}_t)\|^2 \leq F(\mathbf{x}_1) - F(\mathbf{x}_{t+1})
$$

Due to the fact that $0 \leq R_t \leq N$ for all $t$ and $F(\mathbf{x}^*) \leq F(\mathbf{x}_{t+1})$, in the case where $R_t \leq \mathcal{P}_N^t/2$, the algorithm does not converge.

With $\|\Delta_t\|_2 \leq \eta_s \mathcal{L}_N^t$ and $\nabla F(\mathbf{x}^t) = \eta_s \mathcal{L}_1^t$, we have $\mathcal{P}_N^t = \frac{\|\Delta_t\|_2}{\eta_s \mathcal{L}_1^t} \leq \frac{\mathcal{L}_N}{\mathcal{L}_1} \leq \frac{1-(1-c)^N}{c}$ for all $t < T$. In the best case where $R_t = N$, the convergence rate can be derived as follows:

$$\frac{1}{T} \sum_{t=1}^{T} \|\nabla F(\mathbf{x}_t)\|^2 \leq \frac{F(\mathbf{x}_1) - F(\mathbf{x}^*)(\sqrt{\beta_2}G + \epsilon)}{(\frac{Nc}{1-(1-c)^N} - \frac{1}{2})\eta T}$$

When $N = 1$, the convergence rate of $N\text{-AccAdam}$ is the same as $\text{Adam}$ (Zaheer et al., 2018).

### A.2  Proof for Theorem 2

In this section we provide the proof for Theorem 2. We analyze the convergence with no momentum $(\beta_1 = 0)$ and $\alpha = 0$ here. Similar to the proof for Theorem 1, the convergence analysis can be extended to general cases. The notation in the proof follows A.1. In $\text{AdaFedAdam}$, $\mathbf{g}_t$ is given by $\mathbf{g}_t := \frac{\sum p_k \mathbf{U}_k}{\sum p_k}$ where $\mathbf{U}_k$ is the normalized local update given by client $k$ in round $t$ with its certainty $C_k$ (i.e. $\|\mathbf{U}_k\|_2 = \|\nabla_k\|_2$ and $\Delta_k = \eta_k C_k \mathbf{U}_k$). The certainty of $\mathbf{g}_t$ is given by $C_t := \sqrt{\sum p_k C_k}$.

Recall that the update rule of $\text{AdaFedAdam}$ is given by

$$\mathbf{x}_{t+1} = \mathbf{x}_t - (\log C_t + 1)\eta_0 \frac{\mathbf{g}_t}{\sqrt{v_t} + \epsilon}$$

for all $i \in [d]$. Let $R_t := \min \|\frac{\mathbf{g}_{t,i}}{\nabla F(\mathbf{x}_t)_i}\|$ for $i \in [d]$. L-smoothness of the function $F(\mathbf{x})$ guarantees that

$$
\begin{aligned}
F(\mathbf{x}_{t+1}) \leq & F(\mathbf{x}_t) + \langle \nabla_t, \mathbf{x}_{t+1} - \mathbf{x}_t \rangle + \frac{L}{2}\|\mathbf{x}_{t+1} - \mathbf{x}_t\|_2^2 \\
= & F(\mathbf{x}_t) - (\log C_t + 1)\eta_0 \sum_{i=1}^{d} (\nabla_{t,i} \cdot \frac{\mathbf{g}_{t,i}}{\sqrt{v_{t,i}} + \epsilon}) + \frac{L((\log C_t + 1)\eta_0)^2}{2} \sum_{i=1}^{d} \frac{\mathbf{g}_{t,i}^2}{(\sqrt{v_{t,i}} + \epsilon)^2} \\
= & F(\mathbf{x}_t) - (\log C_t + 1)\eta_0 \sum_{i=1}^{d} (\nabla_{t,i} \cdot (\frac{\mathbf{g}_{t,i}}{\sqrt{v_{t,i}} + \epsilon} - \frac{R_t \nabla_{t,i}}{\sqrt{\beta_{2,t} v_{t-1,i}} + \epsilon} + \frac{R_t \nabla_{t,i}}{\sqrt{\beta_{2,t} v_{t-1,i}} + \epsilon})) \\
& + \frac{L((\log C_t + 1)\eta_0)^2}{2} \sum_{i=1}^{d} \frac{\mathbf{g}_{t,i}^2}{(\sqrt{v_{t,i}} + \epsilon)^2} \\
\leq & F(\mathbf{x}_t) - R_t(\log C_t + 1)\eta_0 \sum_{i=1}^{d} \frac{\nabla_{t,i}^2}{\sqrt{\beta_{2,t} v_{t-1,i}} + \epsilon} \\
& + (\log C_t + 1)\eta_0 \sum_{i=1}^{d} \nabla_{t,i} \underbrace{|\frac{\mathbf{g}_{t,i}}{\sqrt{v_{t,i}} + \epsilon} - \frac{R_t \nabla_{t,i}}{\sqrt{\beta_{2,t} v_{t-1,i}} + \epsilon}|}_{T} \\
& + \frac{L((\log C_t + 1)\eta_0)^2}{2} \sum_{i=1}^{d} \frac{\mathbf{g}_{t,i}^2}{(\sqrt{v_{t,i}} + \epsilon)^2}
\end{aligned}
$$

$T$ is bounded by

$$
\begin{aligned}
T &= \left| \frac{\mathbf{g}_{t,i}}{\sqrt{v_{t,i}} + \epsilon} - \frac{R_t \nabla_{t,i}}{\sqrt{\beta_{2,t} v_{t-1,i}} + \epsilon} \right| \\
&\leq \left| \frac{\mathbf{g}_{t,i}}{\sqrt{v_{t,i}} + \epsilon} - \frac{\mathbf{g}_{t,i}}{\sqrt{\beta_{2,t} v_{t-1,i}} + \epsilon} \right| \\
&\leq |\mathbf{g}_{t,i}| \cdot \left| \frac{1}{\sqrt{v_{t,i}} + \epsilon} - \frac{1}{\sqrt{\beta_{2,t} v_{t-1,i}} + \epsilon} \right| \\
&= \frac{|\mathbf{g}_{t,i}|}{(\sqrt{v_{t,i}} + \epsilon)(\sqrt{\beta_{2,t} v_{t-1,i}} + \epsilon)} \cdot \frac{(1 - \beta_{2,t})\mathbf{g}_{t,i}^2}{\sqrt{v_{t,i}} + \sqrt{\beta_{2,t} v_{t-1,i}}} \\
&\leq \frac{1}{(\sqrt{v_{t,i}} + \epsilon)(\sqrt{\beta_{2,t} v_{t-1,i}} + \epsilon)} \cdot \sqrt{1 - \beta_{2,t}} \mathbf{g}_{t,i}^2 \\
&\leq \frac{\sqrt{1 - \beta_{2,t}} \mathbf{g}_{t,i}^2}{(\sqrt{\beta_{2,t} v_{t-1,i}} + \epsilon)\epsilon}
\end{aligned}
$$

With the bound above and $\|\nabla F(\mathbf{x}_t)\|_\infty \leq G$, we have following

$$
\begin{aligned}
F(\mathbf{x}_{t+1}) \leq &F(\mathbf{x}_t) - R_t(\log C_t + 1)\eta_0 \sum_{i=1}^{d} \frac{\nabla_{t,i}^2}{\sqrt{\beta_{2,t} v_{t-1,i}} + \epsilon} \\
&+ \frac{(\log C_t + 1)\eta_0 G\sqrt{1 - \beta_{2,t}}}{\epsilon} \sum_{i=1}^{d} \frac{\mathbf{g}_{t,i}^2}{\sqrt{\beta_{2,t} v_{t-1,i}} + \epsilon} \\
&+ \frac{L((\log C_t + 1)\eta_0)^2}{2\epsilon} \sum_{i=1}^{d} \frac{\mathbf{g}_{t,i}^2}{\sqrt{v_{t,i}} + \epsilon} \\
\leq &F(\mathbf{x}_t) - (\log C_t + 1)\eta_0 R_t \sum_{i=1}^{d} \frac{\nabla_{t,i}^2}{\sqrt{\beta_{2,t} v_{t-1,i}} + \epsilon} \\
&+ \frac{(\log C_t + 1)\eta_0 G\sqrt{1 - \beta_{2,t}}}{\epsilon} \sum_{i=1}^{d} \frac{\nabla_{t,i}^2}{\sqrt{\beta_{2,t} v_{t-1,i}} + \epsilon} \\
&+ \frac{L((\log C_t + 1)\eta_0)^2}{2\epsilon} \sum_{i=1}^{d} \frac{\nabla_{t,i}^2}{\sqrt{\beta_{2,t} v_{t-1,i}} + \epsilon}
\end{aligned}
$$

From the parameters $\eta, \epsilon$ and $\beta$ stated in `Adam`, $L\eta_0/2\epsilon \leq 1/4$ and $G\sqrt{1 - \beta_{2,0}}/\epsilon \leq 1/4$ hold. The inequality $G\sqrt{1 - \beta_{2,0}^{C_t}}/\eta \leq (\log C_t + 1)/4$ holds if $\beta_{2,0} \geq \log^2 2 \approx 0.520$ and $C_t \geq 1$, which is true since $\beta_{2,0}$ is close to 1 with the default value $\beta_{2,0} = 0.999$ and $C_t \geq 1$. Using the inequality conditions, we have

$$
\begin{aligned}
F(\mathbf{x}_{t+1}) &\leq F(\mathbf{x}_t) - (R_t - \frac{(\log C_t + 1)}{2})(\log C_t + 1)\eta_0 \sum_{i=1}^{d} \frac{\nabla_{t,i}^2}{\sqrt{\beta_{2,t} v_{t-1,i}} + \epsilon} \\
&\leq F(\mathbf{x}_t) - \frac{R_t}{2} \frac{(\log C_t + 1)\eta_0}{\sqrt{\beta_{2,0}} G + \epsilon} \|\nabla F(\mathbf{x}_t)\|^2
\end{aligned}
$$

The second inequality is due to the fact that $R_t \geq C_t > (\log C_t + 1)$ and $\beta_{2,t} \leq \beta_{2,0}$ if $C_t \geq 1$. Using a telescope sum and rearranging the inequality, we have

$$
\frac{\eta_0}{\sqrt{\beta_{2,0}} G + \epsilon} \sum_{t=1}^{T} R_t(\log C_t + 1)\|\nabla F(\mathbf{x}_t)\|^2 \leq F(\mathbf{x}_1) - F(\mathbf{x}_{t+1})
$$

Let $R := \min R_t$ and $C := \min C_t$ for all $t \in [T]$, by rearranging the inequality, we obtain

$$\frac{1}{T}\sum_{t=1}^{T}\|\nabla F(\mathbf{x}_t)\|^2 \leq \frac{2(F(\mathbf{x}_1) - F(\mathbf{x}^*))(\sqrt{\beta_{2,0}}G + \epsilon)}{R(\log C + 1)\eta_0 T}$$

# B   PSEUDO CODES FOR ALGORITHMS

Pseudo codes for `Adam`, $N$-`AccAdam` and minibatch `SGD` are given as Algorithm 3 and Algorithm 4.

---

**Algorithm 3** `Adam` and $N$-`AccAdam`

---

**Require:** model weights $\mathbf{x}^0$, stepsize $\eta$, $\beta_1$, $\beta_2$, $\epsilon$ for both `Adam` and $N$-`AccAdam`, $\eta_s$ and $N$ for $N$-`AccAdam`

    $m_0 \leftarrow 0$
    $v_0 \leftarrow 0$
    **for** step $t$ in $\{0, 1, ...T-1\}$ **do**
        `Adam`: $\Delta_t = \nabla_{\zeta \sim \mathcal{D}} F(\mathbf{x}^t)$
        $N$-`AccAdam`: $\Delta_t = (\mathbf{x}^t - \mathbf{SGD}(\mathbf{x}^t, \nabla_{\zeta \sim \mathcal{D}}, \eta_s, N))/\eta_s$
        $m_{t+1} \leftarrow (1 - \beta_1)\Delta_t + \beta_1 m_t$
        $v_{t+1} \leftarrow (1 - \beta_2)\Delta_t^2 + \beta_2 v_t$
        $\hat{m}_{t+1} \leftarrow m_{t+1}/(1 - \beta_1^{t+1})$
        $\hat{v}_{t+1} \leftarrow v_{t+1}/(1 - \beta_2^{t+1})$
        $\mathbf{x}^{t+1} \leftarrow \mathbf{x}^t - \eta \hat{m}_{t+1}/(\sqrt{\hat{v}_{t+1}} + \epsilon)$
    **end for**

---

---

**Algorithm 4** Minibatch Stochastic Gradient Descent (`SGD`)

---

**Require:** model weights $\mathbf{x}^0$, learning rate $\eta_s$, batch size $\zeta$ and number of steps $N$

    **for** step $n$ in $\{0, 1, ...N-1\}$ **do**
        Sample a batch of data with size of $\zeta$ from the training dataset
        Calculate gradient estimation $\nabla_{\zeta \sim \mathcal{D}} F(\mathbf{x}^n)$
        Update model $\mathbf{x}^{n+1} = \mathbf{x}^n - \eta_s \nabla_{\zeta \sim \mathcal{D}} F(\mathbf{x}^n)$
    **end for**

---

Pseudo codes for `FedOpt` (Reddi et al. (2020)) is given as Algorithm 5.

---

**Algorithm 5** Adaptive federated optimization (`FedOpt`)

---

**Require:** Seed model $\mathbf{x}^0$

    **for** round $t$ in $\{0, 1, ...T-1\}$ **do**
        **for** client $k$ in $\{0, 1, ...K-1\}$ **parallel do**
            $\mathbf{x}_k^t := \mathbf{ClientOpt}(\mathbf{x}^t)$                  ▷ Client-side
            $\Delta_k^t := \mathbf{x}_k^t - \mathbf{x}^t$
        **end for**
        $\Delta^t := \mathbf{Aggre}(\{\Delta_k^t, 0 \leq k < K\})$             ▷ Server-side
        $\mathbf{x}^{t+1} := \mathbf{ServerOpt}(\Delta^t)$
    **end for**

---

## C  Experiments

### C.1  Experimental details

**Platform**  All experiments in the paper are conducted on a server with Intel(R) Xeon(R) Gold 6230R CPU and and 2x NVidia RTX A5000 GPUs. All codes are implemented in *PyTorch.*

**Setups**  Details of all federated setups are shown as follows:

- **Femnist:** A multi-layer perceptron network (MLP) for the classification of the EMNIST dataset. The MLP used for the setup consisted 128 hidden nodes activated by ReLu functions with a loss function of cross-entropy. The EMNIST dataset is partitioned according to the writer of images and each partition acts as a local dataset for each client. Local datasets are thus intrinsically non-IID due to different writing characteristics from different writers.

- **Cifar10:** A VGG11 (Simonyan & Zisserman (2014)) model for Cifar10 dataset. The model used for the setup is VGG11 with slight modifications to be compatible with Cifar10 dataset. The architecture of the model is shown as Figure 4 with a loss function of cross-entropy. The Cifar10 dataset is partitioned into 16 subsets by the Dirichlet distribution $\mathbf{Dir}_{16}(0.1)$ over labels.

- **Sent140:** An LSTM model (Gers et al. (2000)) for the sentiment analysis for the Sent140 dataset (Go et al. (2009)). Input words are embedded with pretrained Glove (Pennington et al. (2014)) and logits are output after two LSTM layers with 100 hidden units and one dense layer, with architecture shown in Figure 5. The partitioning of the Sent140 dataset follows Caldas et al. (2018) and a collection of tweets from each twitter account acts as the local dataset of one client.

- **Synthetic:** A linear regression classifier for multi-class classification on a synthetic dataset, proposed by Caldas et al. (2018) as a challenging task for the benchmark of federated learning algorithms. The model is $y = \mathrm{argmax}(\mathrm{softmax}(\mathbf{W}x + b))$, where $x \in \mathbb{R}^{60}$, $\mathbf{W} \in \mathbb{R}^{10 \times 60}$ and $b \in \mathbb{R}^{10}$ with a loss function of cross-entropy. In the Synthetic dataset, there are 100 partitions, the sizes of which follow a power law.

For all setups, each client is associated with a partition and randomly split the local partition with a ratio of $8 : 2$ acting as its local training and testing set before federated training starts. A summary of four setups are shown in Table 4.

Table 4: A summary of setups: the four setups cover different Federated Learning scenarios, non-IID types and task types.

| Setup | # Clients | Model | Scenario | Non-IID Type | Task Type |
|---|---|---|---|---|---|
| Femnist | 3500 | MLP | Cross Device | Intrinsic | Computer Vision |
| Cifar10 | 16 | CNN | Cross Silo | Dirichlet | Computer Vision |
| Sent140 | 697 | LSTM | Cross Device | Intrinsic | Natural Language Process |
| Synthetic | 100 | Linear Model | Cross Device/Silo | Synthetic | Classification |

**Hyperparameter settings**  Hyperparemters of federated optimization methods can be categorized as a). method-independent hyperparameters (including local learning rate, batch size, number of local epochs and communication rounds), and b). method-specific hyperparameters (e.g. $q$ in q-FedAvg, $\mu$ in FedProx). It is prohibitively infeasible to find the optimal combinations of hyperparameter of all optimization with grid-search. In this work, we first tune the local learning rate, batch size and number of local epochs on FedAvg, and fix them for other algorithms first. Then the method-specific hyperparameters for individual federated method are tuned to obtain their optimal performance. For all experiments without specifications, local optimizers for clients are fixed as `SGD` and number of local epoch is fixed as 1. Local learning rate for Femnist, Cifar10, Sent140 and Synthetic setup are $0.01, 0.02, 0.3$ and $0.01$, respectively. Batch sizes for

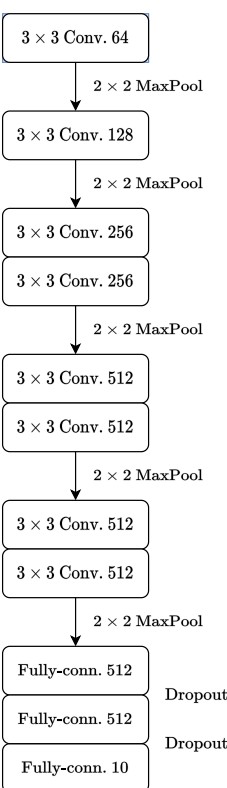

Figure 4: Architecture of VGG11 for the CIFAR10 setup.

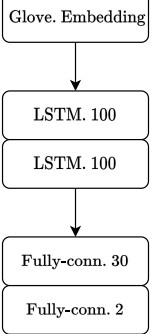

Figure 5: Architecture of the two-layer LSTM for the Sent140 setup.

five setups are $10, 32, 10$ and $10$ respectively. If local optimizers are set as `SGD` with (Nesterov) momentum, the momentum factor is fixed as 0.9 by default. For server optimizers, `FedAvg` has the default learning rate $\eta = 1$, `FedAdam` has the default hyperparameter set ($\eta = 0.001$, $\beta_1 = 0.9$ and $\beta_2 = 0.999$), and `q-FedAvg` has learning rate $\eta = 1$. For all experiments with `q-FedAvg`, $q = 1$ is fixed to compare with `AdaFedAdam`with $\alpha = 1$. For all experiments with `FedProx`, $\mu$ is tuned with grid search from $\{0.001, 0.005, 0.01, 0.1, 1\}$. Total communication rounds are 500, 200, 1000 and 1000 for the Femnist, Cifar10, Sent140 and Synthetic setup, respectively. For each experiment, global models are initialized with 3 different random seeds and trained independently, and averaged metrics are reported.

## C.2 Full experimental results

**Convergence & Fairness**  Table 5 shows the full results of the experiment of fairness and convergence.

Table 5: Full experimental results of convergence and fairness: Statistics of test accuracy on clients for `AdaFedAdam` compared to `FedAvg`, `FedAdam`, `FedNova` and `q-FedAvg` with $q = 1$, for Femnist, Cifar10, Sent140 and Synthetic setups.

| Settings | Algorithms | Avg.(%) | STD.(%) | Worst 30%(%) |
|---|---|---|---|---|
| Femnist | FedAvg | 77.77 ±0.64 | 13.2 ±0.91 | 60.11 ±2.19 |
| | FedAdam | 82.97 ±0.26 | 11.44 ±0.76 | 67.65 ±1.8 |
| | q-FedAvg | 76.91 ±0.21 | 10.94 ±0.26 | 64.06 ±0.37 |
| | FedNova | 78.31 ±0.53 | 10.77 ±0.36 | 64.97 ±1.01 |
| | FedProx | 80.16 ±0.16 | 10.02 ±0.19 | 66.3 ±0.38 |
| | AdaFedAdam | 84.48 ±0.5 | 8.62 ±0.25 | 74.16 ±0.3 |
| Cifar10 | FedAvg | 48.3 ±0.37 | 14.33 ±1.3 | 26.43 ±1.09 |
| | FedAdam | 59.35 ±1.43 | 7.74 ±0.59 | 49.9 ±0.7 |
| | q-FedAvg | 31.94 ±1.09 | 13.49 ±0.05 | 13.18 ±1.37 |
| | FedNova | 40.32 ±2.01 | 18.99 ±1.65 | 20.06 ±3.31 |
| | FedProx | 55.77 ±0.28 | 13.73 ±0.94 | 39.11 ±2.66 |
| | AdaFedAdam | 65.27 ±1.32 | 7.65 ±1.62 | 57.12 ±1.61 |
| Sent140 | FedAvg | 73.4 ±1.02 | 18.76 ±1.08 | 49.82 ±2.75 |
| | FedAdam | 73.82 ±0.91 | 15.47 ±0.37 | 55.54 ±0.23 |
| | q-FedAvg | 73.38 ±0.61 | 16.99 ±1.02 | 52.88 ±2.08 |
| | FedNova | 74.64 ±0.51 | 17.6 ±0.95 | 53.03 ±1.86 |
| | FedProx | 74.72 ±0.42 | 16.74 ±1.29 | 54.65 ±2.32 |
| | AdaFedAdam | 74.62 ±0.07 | 10.24 ±0.35 | 58.8 ±0.33 |
| Synthetic | FedAvg | 90.08 ±0.12 | 14.23 ±0.28 | 39.51 ±1.9 |
| | FedAdam | 89.97 ±0.16 | 13.52 ±0.29 | 53.08 ±2.95 |
| | q-FedAvg | 90.64 ±0.1 | 10.36 ±0.16 | 78.65 ±0.23 |
| | FedNova | 92.97 ±0.11 | 8.06 ±0.11 | 83.16 ±0.3 |
| | FedProx | 91.74 ±0.06 | 15.53 ±0.07 | 28.62 ±0.3 |
| | AdaFedAdam | 95.07 ±0.13 | 5.5 ±0.2 | 88.64 ±0.42 |

**Robustness against different levels of data heterogeneity**  Figure 6 shows label distributions of different non-IID levels of the Cifar10 setup. Table 6 shows the full results of comparison between different algorithms on the Cifar10 setup. Different levels of data heterogeneity are generated with Dirichlet distribution of different concentration parameter $\beta$ ranging from 0.05 to 1 and together with an IID partitioning. It can be observed that `AdaFedAdam` consistently outperforms other algorithms with the highest test accuracy and lowest STD of test accuracy in all different settings.

**Compatibility with local momentum**  Table 7 shows the full results of different federated algorithms with different local solvers. It is observed that `AdaFedAdam` is not only compatible with momentum-based local solvers, it also provides better results compared to other federated algorithms.

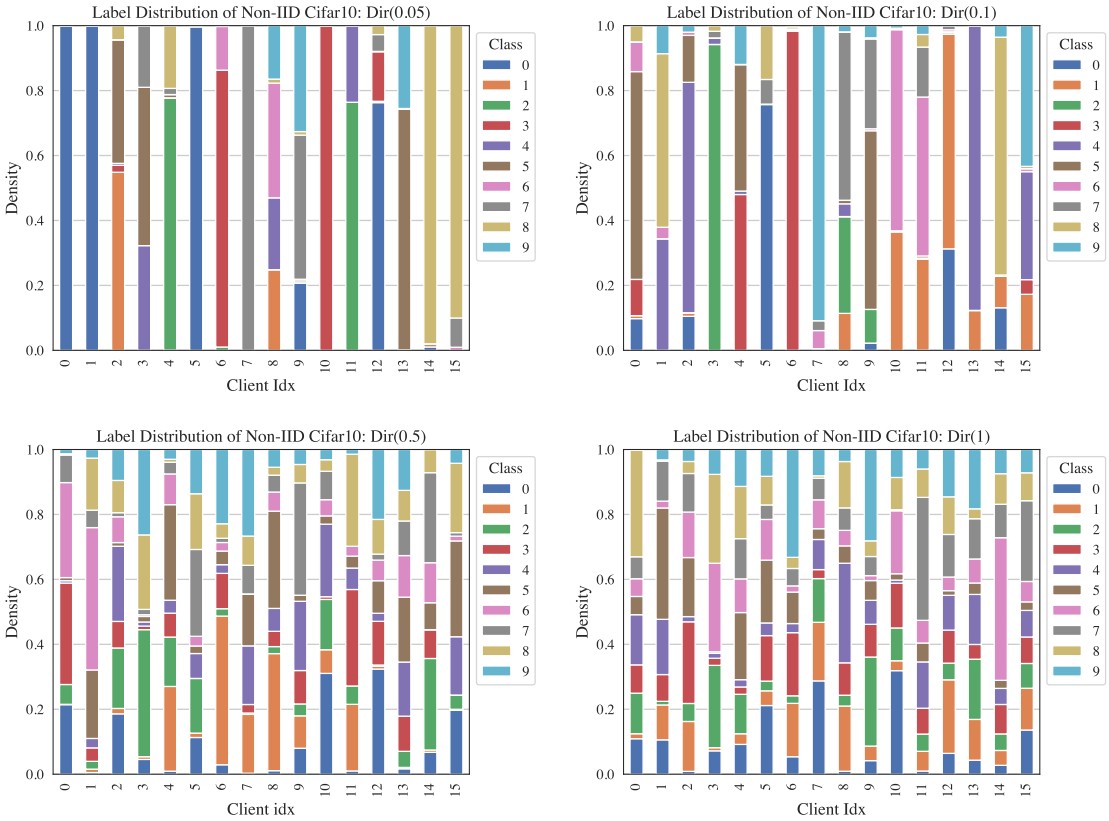

Figure 6: Label distributions of local datasets of the Cifar10 setup for different non-IID levels.

Table 6: Full experimental results of federated algorithms against different levels of data heterogeneity on the Cifar10 setup.

| Data Distribution | Algorithm | Avg.(%) | STD.(%) | Worst 30%(%) |
|---|---|---|---|---|
| **Dir**(0.05) | FedAvg | 36.47 ±0.75 | 20.28 ±0.90 | 9.45 ±3.51 |
| | FedAdam | 56.33 ±0.96 | 11.77 ±1.99 | 40.4 ±5.13 |
| | q-FedAvg | 28.01 ±0.81 | 21.92 ±0.47 | 3.15 ±3.25 |
| | FedNova | 36.25 ±0.88 | 24.34 ±1.34 | 5.00 ±3.24 |
| | AdaFedAdam | 62.81 ±1.02 | 8.18 ±1.33 | 46.01 ±2.23 |
| **Dir**(0.1) | FedAvg | 50.41 ±0.46 | 13.21 ±0.36 | 33.20 ±3.98 |
| | FedAdam | 65.79 ±0.91 | 8.61 ±0.51 | 55.92 ±2.36 |
| | q-FedAvg | 38.95 ±0.73 | 12.46 ±0.20 | 24.59 ±2.11 |
| | FedNova | 48.09 ±1.82 | 14.29 ±0.58 | 33.34 ±3.28 |
| | AdaFedAdam | 66.16 ±1.13 | 8.59 ±0.39 | 56.48 ±1.45 |
| **Dir**(0.5) | FedAvg | 49.38 ±0.92 | 7.29 ±2.60 | 41.22 ±3.25 |
| | FedAdam | 70.49 ±0.78 | 3.97 ±0.49 | 65.83 ±0.84 |
| | q-FedAvg | 44.95 ±0.41 | 4.60 ±0.51 | 39.73 ±0.61 |
| | FedNova | 49.47 ±1.06 | 6.09 ±2.19 | 43.00 ±3.26 |
| | AdaFedAdam | 71.43 ±0.81 | 5.40 ±0.22 | 64.93 ±1.04 |
| **Dir**(1): | FedAvg | 40.97 ±0.66 | 4.93 ±0.47 | 35.53 ±1.12 |
| | FedAdam | 71.22 ±0.17 | 2.95 ±0.27 | 68.01 ±0.02 |
| | q-FedAvg | 36.27 ±0.95 | 5.40 ±0.75 | 30.63 ±1.56 |
| | FedNova | 40.28 ±0.10 | 4.70 ±0.49 | 35.30 ±0.40 |
| | AdaFedAdam | 72.77 ±0.44 | 3.05 ±0.11 | 69.57 ±0.54 |

Table 7: Full experimental results of federated algorithms in cooperation with local momentum on the Synthetic setup.

| Local Solver | Algorithm | Avg.(%) | STD.(%) | Worst 30%(%) |
|---|---|---|---|---|
| Vanilla `SGD` | `FedAvg` | 90.08 ±0.12 | 14.23 ±0.28 | 39.51 ±1.9 |
| | `FedAdam` | 89.97 ±0.16 | 13.52 ±0.29 | 53.08 ±2.95 |
| | `q-FedAvg` | 90.64 ±0.1 | 10.36 ±0.16 | 78.65 ±0.23 |
| | `FedNova` | 92.97 ±0.11 | 8.06 ±0.11 | 83.16 ±0.3 |
| | `FedProx` | 91.74 ±0.06 | 15.53 ±0.07 | 28.62 ±0.3 |
| | `AdaFedAdam` | 95.07 ±0.13 | 5.5 ±0.2 | 88.64 ±0.42 |
| SGD with Momen. | `FedAvg` | 95.26 ±0.22 | 8.42 ±0.24 | 68.65 ±0.19 |
| | `FedAdam` | 91.60 ±0.32 | 12.32 ±0.66 | 59.52 ±4.79 |
| | `q-FedAvg` | 94.64 ±0.22 | 5.73 ±0.03 | 88.04 ±0.02 |
| | `FedNova` | 96.12 ±0.09 | 3.82 ±0.05 | 93.07 ±0.15 |
| | `FedProx` | 93.12 ±0.06 | 7.21 ±0.05 | 84.53 ±0.09 |
| | `AdaFedAdam` | 97.19 ±0.11 | 3.32 ±0.02 | 93.41 ±0.11 |
| SGD with Neste. Momen. | `FedAvg` | 95.24 ±0.14 | 8.34 ±0.05 | 68.80 ±0.36 |
| | `FedAdam` | 91.79 ±0.19 | 12.07 ±0.43 | 61.51 ±3.52 |
| | `q-FedAvg` | 94.56 ±0.02 | 5.80 ±0.12 | 87.85 ±0.02 |
| | `FedNova` | 96.85 ±0.04 | 3.80 ±0.30 | 94.02 ±0.11 |
| | `FedProx` | 95.46 ±0.15 | 5.37 ±0.12 | 87.41 ±0.93 |
| | `AdaFedAdam` | 97.27 ±0.16 | 3.19 ±0.21 | 94.19 ±0.24 |

