# OpenReview forum: "Accelerating Fair Federated Learning: Adaptive Federated Adam"
_TMLR — Rejected by TMLR_

### Review · Reviewer_gbcq · 2023-06-01

**Summary Of Contributions:**

1. This paper proposes a new distributed optimizer for federated learning AdaFedAdam, with better convergence and fairness than existing methods. For optimization goal, AdaFedAdam shares the same fairness notion as $q$-fair federated learning. However, AdaFedAdam incorporates an additional learning rate weighting scheme by leveraging running statistics (e.g., gradient norm) across different agents, which alleviates the hyperparameter tuning burden and improves fairness and convergence.
2. A theoretical convergence guarantee is derived.
3. Extensive experiments on multiple datasets show the practical significance of the proposed method.

**Audience:**

Yes

**Broader Impact Concerns:**

The limitation and scope of the considered fairness notion may need to be discussed. For example, this fairness notion mainly cares about equalized performance measured by loss across different agents. The notion does not imply universal fairness, e.g., fairness among different subgroups within the data distribution.

**Claims And Evidence:**

Yes

**Requested Changes:**

Writing issues:
1. After Eqn. (5), what does "where identical notations in equation 1" mean?
2. After Eqn. (7), $\alpha$-Fairn -> $\alpha$-Fair
3. Last but one paragraph of Section 4.1, "Low certainty ... also only allow" -> "Low certainty ... also only allows"
4. On page 6, the normalization of local updates paragraph has a few notations appearing only in Alg 2, such as $U_k$, $Delta_k$, etc. But Alg 2 is on page 7. So it isn't very clear when reading this paragraph.
5. Eqn. (9), additional ")" in LHS.
6. On Page 10, "Table 6" should be "Table 1".
7. In Table 1, please note that the metric is accuracy.
8. Missing period in the last paragraph of Section 6.

Question on experimental evaluation:
Could you compare AdaFedAdam of different alpha values with the corresponding qfedavg? If I understand correctly, the alpha in AdaFedAdam and q in qfedavg are inspired by the same fairness notion so they could be comparable.


**Strengths And Weaknesses:**

Strengths:
1. Identified and analyzed the fairness and hyperparameter tuning issue of FedAdam, a widely-used federated learning optimizer.
2. A well-motivated method to improve fairness and convergence in FL following the identified fairness and hyperparameter tuning issue.
3. Strong empirical performance across multiple practical datasets and comprehensive experimental settings.

Weaknesses:
1. Though the theoretical convergence guarantee is given, no theoretical benefit could be observed when comparing the bound with the bounds of existing methods. So the theoretical contribution is a bit limited.
2. The writing quality could be further improved.

---

> ### Author Response · Authors · 2023-06-20
>
> We are grateful for the time and efforts you have invested in reviewing our work. We hope that our response will address your concerns.
>
> - Experimental evaluation: We acknowledge and agree with the reviewer's point regarding the feasibility and importance of comparing the parameter $q$ in q-FedAvg and the parameter $\alpha$ in AdaFedAdam. In response to this comment, we have updated Figure 3 in the manuscript to include q-FedAvg with $q\in\[1, 2, 4\]$. It is observed that AdaFedAdam consistently outperforms the corresponding q-FedAvg algorithm in terms of both fairness and convergence across different values of $\alpha$ and $q$.
>
> - Notion of fairness: The term *fairness* encompasses various interpretations such as group fairness and performance fairness in the field of machine learning [1]. In our paper, we specifically employ the term *fairness* to describe disparities in model performance observed among participants in a federated learning process, following the convention of previous studies [2, 3]. To clarify any ambiguities, we have placed significant emphasis on clearly defining the term *fairness* in the revised manuscript.
>
> - Writing issues: We apologize for the presence of typos in the manuscript and express our gratitude for bringing them to our attention. We have addressed these typos and carefully reviewed the manuscript to ensure clarity and coherence.
> ---
> References:
>
> [1] Mehrabi, Ninareh, et al. "A survey on bias and fairness in machine learning." _ACM Computing Surveys (CSUR)_ 54.6 (2021): 1-35.
>
> [2] Mohri, Mehryar, Gary Sivek, and Ananda Theertha Suresh. "Agnostic federated learning." _International Conference on Machine Learning_. PMLR, 2019.
>
> [3] Li, Tian, et al. "Fair resource allocation in federated learning." _arXiv preprint arXiv:1905.10497_ (2019).

---

### Review · Reviewer_rvFp · 2023-06-04

**Summary Of Contributions:**

The paper studies the issue of fairness in Federated Learning and proposes a novel approach to tackle it. The authors introduce a dynamic formulation of the adjusted empirical risk minimization problem to enforce fairness. They develop a new adaptive method called AdaFedAdam, inspired by Federated Adam, which dynamically adjusts the optimizer's parameters during the training process. The paper includes a convergence analysis of AdaFedAdam and Adam with Accumulated updates. Experimental results demonstrate that the suggested method outperforms several benchmark approaches in terms of faster convergence and improved fairness across four different tasks.

**Audience:**

Yes

**Broader Impact Concerns:**

No concerns from my side for work of such type.

**Claims And Evidence:**

No

**Requested Changes:**

In my opinion, the paper requires significant revision due to numerous typos and unconventional wording choices that hinder readability and comprehension. Many sections suffer from imprecision and confusion, making it difficult to discern the intended meaning. As a result, the paper gives the impression of being a draft rather than a polished submission.

**Strengths And Weaknesses:**

# Strengths

The research direction pursued by the authors appears to be innovative and relatively unexplored in the existing literature. The problem of convergence degradation caused by fairness constraints is crucial for the practical implementation of federated learning systems, as efficiency is a significant concern in federated training. Furthermore, exploring adaptive hyperparameter tuning is a valuable and understudied aspect within the realm of federated learning.


# Weaknesses

1. **Problem formulation.**
In my opinion, the justification for the chosen problem formulation (2) is not convincing. It remains to be seen why this specific approach to fairness, which is also not formally defined, was selected and how it compares to alternative methods. Moreover, the adjustment of $\alpha$-Fair FL seems problematic as it contains $I_k^\alpha(t)$ and implicitly depends on the method's iterates $x^t$, making it unclear how the loss (and gradient) can be computed without running the optimization procedure.

In addition, derivations and arguments in Subsection 3.3 need more rigor and formalization. An assumption like $F_i(x^0) \simeq F_j(x^0)$ does not seem reasonable and can be violated for simple quadratic losses $F_i(x) = x^\top A_i x - b_i^\top x + c_i$. The gradient calculation is incomprehensible to me, and the sign $\simeq$ used in Equation (6) is undefined. The proposed problem formulation is claimed to have fairness properties, but these statements are not proven. It is also unclear what they even mean: do they refer to the optimal solution of the problem formulation (5)? I would expect that, in practice, optimization error has to be taken into account as well.

2. **Convergence Analysis.** The assumption of bounded gradients presents a significant issue as it limits heterogeneity and has been shown to be "pathological" for the analysis of distributed optimization methods [1].
The convergence upper bounds provided may not be informative as they do not decrease with the number of iterations, potentially indicating a typo in the right-hand sides of the results. Furthermore, terms like $R$ and $C^t$ in the bounds have obscure effects on algorithms convergence. It is unclear whether $R$ is even bounded.

Formulation of Theorem 1 includes the statement:

> *SGD exhibits approximately linear convergence*

which is not thoroughly explained in the mean text and seems incorrect, as, in the general finite-sum case, SGD converges linearly  to the neighborhood of the solution [4].

In Subsection 4.2, it is said that
> *When data between clients are statistically homogeneous, the average of local updates is an unbiased estimator of accumulated updates of multiple centralized SGD steps.*

which is technically incorrect (or needs an explanation of what is meant by *"statistically homogeneous"*). It is known [2, 3] that FedAvg-type methods, including local Gradient Descent, have biased gradient estimators and converge to different stationary point rather than the Empirical Risk Minimizer of the original problem.

3. **Notation issues.** There are multiple issues with the notation used in the paper. The problem is defined for $F(x)$, but convergence results are presented for $f(x)$. It's unclear if the problem's finite-sum structure is considered. Additionally, the meaning of $\nabla_{\xi \sim \mathcal{D}} F(x)$ is not clearly explained. The definition of $S_k$, used in Algorithm 2, is lacking. Please correct me if I missed it.

4. **Experimental results.**
The experimental setup could be more transparent for other methods like FedAdam, $q$-FedAvg, etc. Namely, what is the decoupling of the problem formulation and optimization method? The impact of the proposed adaptive strategy is not clearly demonstrated. The rationale behind fixing $q=1$ for $q$-FedAvg is not explained.

Why does test accuracy degrade for the Sent140 task in the case of FedAdam and AdaFedAdam? What happens if training continues? The current trend indicates that FedAvg and FedNova may reach the best test accuracy. Why was training stopped at 1000 rounds? The interpretation of the results from the last row in Figure 2 is not apparent. Especially the phrase *“the most uniform distribution”*. Can it be formally and quantitatively characterized?

The design of the method is obscure, with numerous adjustments made to the original FedAdam. The interplay between these adjustments and their impact on performance should be studied through ablation studies.

The term *"Relative Standard Deviation (RSD)"* should be defined to provide clarity and understanding.

___


[1] Khaled, Ahmed, Konstantin Mishchenko, and Peter Richtárik. "Tighter theory for local SGD on identical and heterogeneous data." International Conference on Artificial Intelligence and Statistics. PMLR, 2020.

[2] Malinovskiy, Grigory, et al. "From local SGD to local fixed-point methods for federated learning." International Conference on Machine Learning. PMLR, 2020.

[3] Charles, Zachary, and Jakub Konečný. "On the outsized importance of learning rates in local update methods." arXiv preprint arXiv:2007.00878 (2020).

[4] Gower, Robert Mansel, et al. "SGD: General analysis and improved rates." International conference on machine learning. PMLR, 2019.

---

> ### Author Response · Authors · 2023-06-20
> **Response: Part 1**
>
> We are very grateful for your enthusiastic feedback! Please find below our response on the mentioned issues.
>
> - Problem formulation
>     1. The formulation depends on $x^t$:  The proposed formulation serves the purpose of dynamically re-balancing the local objective functions $F_k(\mathbf{x})$ within the finite-sum structure. This is achieved by adjusting their weights for averaging based on their respective training progress after each global round. The dynamic formulation presented in our work shares a similar motivation with Adaptive Loss Balancing [1], which is a widely used technique in multi-task learning. The goal is to prevent certain tasks from dominating the training process by ensuring a more balanced contribution from all tasks involved.
>     2. Assumption $F_i(x^0) \simeq F_j(x^0)$: This assumption is applicable when the global (neural network) model is initialized with random weights, such as He initialization[2] and Xavier initialization[3]. The assumption may not hold when pre-training is employed to initialize the global model. As our paper primarily focuses on the proposed technique and not on pre-training in federated learning, the fairness guarantee for such cases falls outside the scope of our work. However, we acknowledge the importance of studying the fairness guarantee in scenarios involving pre-training as a valuable direction for future research. Furthermore, in the revised manuscript, we have provided a clarification regarding the usage of the notation $\simeq$.
> - Convergence analysis
>     1. Typo in Theorem 2: We apologize for the typo of missing $T$ in the denominator in Equation 9. Thank you for pointing this out.
>     2. Convergence of SGD: We have updated the statement of the convergence of SGD in Theorem 1.
>     3. Estimation of accumulated centralized SGD steps: Term "statistically homogeneous" denotes the case where distribution of local datasets is identical ($\forall k, i \in {[K]}, \mathcal{D}_k = \mathcal{D}_i$). We argue that it has been proven in prior work ([4, 5]) that in the homogeneous cases, local SGD converges to the same optima as mini-batch SGD, from which in square distance of $\mathcal{O}(H-1)$, where $H$ denotes number of local steps. In our revised manuscript, we have provided further clarification on the usage of the term "statistically homogeneous" and included a discussion on the estimator in homogeneous cases.
> - Notation issues:
>     1. $F(x)$ and $f(x)$: In the problem formulation $F(x)$ is used to denote the risk functions of the finite-sum structure. However, in the theorems, $f(x)$ is used to denote the objective function with specific assumptions in theorems, which may or may not be in the finite-sum structure like $F(x)$. To enhance the readability, made the necessary revisions in the revised manuscript, replacing all instances of $f(x)$ in the theorems with $F(x)$ though.
>     2. $\nabla_{\zeta\sim \mathcal{D}}F(x)$: This is the notation for the gradient estimation of $F(x)$, from a batch of data $\zeta$ sampled from distribution $\mathcal{D}$, as explained in Section 4.1. We have clarified the notation of the gradient estimation in the revised manuscript.
>     3. $S_k$: We used $S_k$ to denote the size of the local dataset $D_k$. To reduce any ambiguities, we now use the relative size of local dataset $p_k := \frac{|D_k|}{\sum_{i=1}^{K} |D_i|}$ in the revised manuscript.

---

> ### Author Response · Authors · 2023-06-20
> **Response: Part 2**
>
> - Experimental results:
>     1. Transparency for baseline algorithms: We would like to highlight that the detailed information regarding all setups and experiments, including the baseline algorithms, has been provided in the appendix. Additionally, as suggested by Review M4id, we have included the detailed fine-tuning method for all experiments in Appendix C.1. These additions aim to ensure transparency and facilitate a comprehensive understanding of the experimental procedures and methodologies.
>     2. Rationale for $q=1$ in qfedavg: Both adafedadam and q-FedAvg are derived from the $\alpha$-fairness function and are comparable when using the same values for $q$ and $\alpha$. Therefore, in all of our experiments, adafedadam and q-FedAvg have a fixed value of $q=1$ and $\alpha=1$. This choice ensures consistency and allows for a fair comparison between the two methods.
>     3. Performance on Sent140 setup: To demonstrate the convergence of the other algorithms, we have provided updated results for 2000 global rounds on the Sent140 setup. It has been observed that the test accuracy of adafedadam starts to degrade after around 200 global rounds. This behavior can be attributed to overfitting. While the other algorithms continue to converge, adafedadam and FedAdam reach a point of convergence and subsequently begin to overfit. This can be attributed to the invariance of their update to the scale of gradients. This observation is further supported by examining the test error as a function of the number of global rounds: while the test error for the other algorithms continues to decrease, adafedadam and FedAdam have already reached their minimal values, and the test error starts to increase thereafter.
>     5. Interpretation of results: We have made clarifications regarding the interpretation of the last row of Figure 2 in the caption. Specifically, we have revised the statement regarding "the most uniform distribution" and instead compared the quantitative measure of the standard deviation of local accuracy. Additionally, we have provided a definition for the term "Relative Standard Deviation" in the main text to enhance clarity and understanding.
> ---
> References:
>
> [1] Chen, Zhao, et al. "Gradnorm: Gradient normalization for adaptive loss balancing in deep multitask networks." _International conference on machine learning_. PMLR, 2018.
>
> [2] He, Kaiming, et al. "Delving deep into rectifiers: Surpassing human-level performance on imagenet classification." _Proceedings of the IEEE international conference on computer vision_. 2015.
>
> [3] Glorot, Xavier, and Yoshua Bengio. "Understanding the difficulty of training deep feedforward neural networks." _Proceedings of the thirteenth international conference on artificial intelligence and statistics_. JMLR Workshop and Conference Proceedings, 2010.
>
> [4] Khaled, Ahmed, Konstantin Mishchenko, and Peter Richtárik. "Tighter theory for local SGD on identical and heterogeneous data." International Conference on Artificial Intelligence and Statistics. PMLR, 2020.
>
> [5] Stich, Sebastian U. "Local SGD converges fast and communicates little." _arXiv preprint arXiv:1805.09767_ (2018).

---

### Review · Reviewer_M4id · 2023-06-08

**Summary Of Contributions:**

In this study, the focus is on examining the issue of fairness in federated learning. The researchers approach the problem by framing it as a multi-task learning problem. To address this challenge, they introduce adaptive FedAdam, a method that dynamically adjusts the hyper-parameters during the Adam update process. Through experimental evaluations, they demonstrate that this approach achieves improved performance compared to existing methods.

**Audience:**

Yes

**Broader Impact Concerns:**

This study primarily focuses on the design of FL algorithms within the existing framework and does not raise any immediate concerns regarding broader impact. The findings and outcomes of this research may not have direct real-world applications at present. Therefore, the potential broader impact of this work is currently limited.

**Claims And Evidence:**

Yes

**Requested Changes:**

1. The paper lacks a clear and thorough explanation of fairness, the DMOP formulation, and the underlying intuition behind the algorithm. Providing a comprehensive illustration of these aspects would greatly enhance the understanding of the proposed research.
2. To ensure fairness in the comparison, it is crucial to evaluate the proposed algorithms against baselines using the optimal hyperparameters specific to each algorithm. This approach will provide a more accurate assessment of the performance of AdaFedAdam and other methods.
3. The inclusion of additional baseline methods would strengthen the evaluation and contribute to a more comprehensive analysis of the proposed algorithms. It is recommended to incorporate a broader range of baselines in order to provide a better comparison.
4. In the experimental evaluation, it is important to include results with modern backbones in addition to the existing ones such as MLPs and VGGs. Incorporating architectures like ResNet and other contemporary models will help establish the effectiveness and applicability of the proposed algorithms across a wider range of network architectures.
5. The evaluation should be expanded to include results with a larger number of clients. While the current experiments consider 16 clients, it is advisable to include scenarios with a higher number of clients, such as 100, to better reflect real-world settings and provide more robust insights.


**Strengths And Weaknesses:**

Strengths:

1. They provide a convergence guarantee for AdaFedAdam.
2. The researchers conduct several ablation studies to demonstrate the robustness of AdaFedAdam.

Weaknesses:
1. The introduction lacks clarity, particularly in explaining the exact definition of fairness in the federated learning (FL) setting. The authors should include more information in the related works and formulation sections, such as illustrating unfair FL settings and distinguishing them from issues like overfitting in imbalanced data distribution among local clients.

2. Equation (6) proposes a formulation of DMOP, which approximately shares gradients with the fairness problem equation (2). However, it is unclear how this formulation contributes to fairness. The standard FL objective equation (1) appears to be a natural DMOP formulation. Therefore, the benefit of DMOP formulation in fair FL needs further explanation.

3. The connection between the DMOP formulation and the design of AdaFedAdam is not clearly explained. The intuition behind AdaFedAdam seems confusing. Additionally, the focus on FedAdam as the specific algorithm raises questions, as FedAdam is just a variant of FedOPT. It would be helpful to know if the adaptive tuning method can be generalized to other FL algorithms.

4. The claim that finetuning-free is a benefit and leads to better performance with the same parameters inherited from the centralized setting is considered unfair. A fairer approach would involve comparing AdaFedAdam with other methods using optimal hyperparameters for each method. The reason is that (1) FL algorithms typically involve additional hyperparameters compared to the centralized setting. (2) In a real FL scenario, performing centralized training first to select hyperparameters is impractical due to privacy concerns and communication constraints.

5. Experimental considerations:
a. Only MLPs and VGGs are considered as backbones for image tasks, while including results with other architectures like ResNet (e.g., CNNs) would enhance the comprehensiveness of the study.
b. The use of only 16 clients for the CIFAR-10 dataset may not represent real-world scenarios adequately. Including performance evaluation with a larger number of clients, such as 100, would provide more realistic insights.
c. Including more baseline methods like FedProx and SCAFFOLD would strengthen the experimental evaluation.

---

> ### Author Response · Authors · 2023-06-20
> **Response: Part 1**
>
> We are very grateful for your valuable review! To address your concerns, please find our response below.
>
> - Clarification of the definition of fairness in Federated Learning: In the revised manuscript, we have highlighted our definition of "fairness" in the introduction section.
>
>
> - Underlying intuition behind the algorithm: We have added the intuitive explanation for the proposed algorithm in Section 5.1, which may help enhance the understanding of adafedadam.
>
>
> - Benefits of DMOP formulation: As outlined in the last paragraph of Section 3.3, the DMOP formulation offers an identical fairness guarantee when compared to the Fair Federated Learning (FFL) formulation. However, an advantageous characteristic of the DMOP formulation is its ability to overcome the inherent issue of diminishing gradient scales in FFL when employing first-order optimization methods, as explained in Section 3.2.
>
>
> - Generalization of the adaptive tuning method to FedOpt: As mentioned in your comment, FedAdam is just one variant of FedOpt that utilizes Adam as the server optimizer. Considering that Adam is essentially a combination of Momentum-SGD (where $\beta_1$ is derived from) and RMSProp (where $\beta_2$ is derived from), our method can also be applied to FedAvg with server momentum and FedRMSProp. Similarly it is feasible to apply our algorithm to other FedOpt methods with adaptive server optimizers, such as FedAdamW and FedYogi. However, due to the constrained time of the rebuttal period, a comprehensive investigation into the performance of this generalized method is not currently feasible. We sincerely appreciate your insightful comment and we agree that further exploration of this direction is a promising avenue for future research.
>
>
> - Fair comparison with other methods:
>     1. Fine-tuning methods: Hyperparameters of federated optimization methods can be categorized into two groups. Firstly, we have method-independent hyperparameters, including local learning rate, batch size, number of local epochs and communication rounds. Secondly, there are method-specific hyperparameters, such as $q$ in q-FedAvg and $\mu$ in FedProx. Exhaustively searching for the optimal combinations of hyperparameters for all optimization methods using grid-search is prohibitively infeasible. Instead, a more practical approach is to initially tune the local learning rate, batch size, and number of local epochs specifically for FedAvg, and subsequently fix them for other algorithms. Then, the method-specific hyperparameters for each individual federated method can be fine-tuned to achieve optimal performance (as demonstrated in [2, 3]). Following this convention, we have updated the experiment results and all hyperparameters are provided in the Appendix.
>
>     2. Weakness 4 reason 2): The proposed fine-tuning-free method adafedadam does not need to run centralized training to find the optimal centralized hyperparameters, before proceeding with federated training. This is made possible due to the invariance of the Adam step size, denoted as $\eta$, to the gradient scale. Even with the default values of $\eta=0.001$, $\beta_1=0.9$, and $\beta_2=0.999$, Adam consistently performs well in centralized training without requiring fine-tuning. Our proposed method, adafedadam, dynamically adapts the values of $\eta$, $\beta_1$ and $\beta_2$ based on the *default* values in Adam, and provides consistently strong performance in various tasks without the need of fine-tuning.

---

> ### Author Response · Authors · 2023-06-20
> **Response: Part 2**
>
> - Experimental considerations:
>     1. Models of more architectures as the backbone: In our experimental setups, we have employed four different models that encompass the most commonly used types of layers, including dense layers, convolutional layers, and LSTM layers. However, we have deliberately excluded ResNet models as the backbone for image tasks. This decision was motivated by the presence of batch normalization (BN) layers within ResNet. Although the impact of BN layers in federated settings has not been thoroughly studied, recent studies have suggested potential negative impacts of BN in federated learning, particularly in highly non-IID cases [4, 5, 6]. To mitigate any ambiguous interactions between BN layers and federated optimization, we have opted for VGG as the backbone for the Cifar10 dataset.
>
>     2. Number of clients of CIFAR10 setup: CIFAR10 setup is the setup to evaluate the performance of adafedadam in cross-silo scenarios. On the other hand, the FEMNIST setup comprises 3,500 clients, while the SENT140 setup involves 697 clients. Both setups are intended to emulate cross-device scenarios in the evaluation of adafedadam.
>     3. Additional baseline algorithms: We have now included an additional baseline algorithm, FedProx, in the experimental results. However, due to the time constraints during the rebuttal period, we were unable to compare the results with SCAFFOLD. Furthermore, it should be noted that SCAFFOLD incurs double the communication cost of a single global round. Hence, it would be unfair to compare SCAFFOLD with other methods without considering the communication cost. If deemed necessary, we can incorporate SCAFFOLD as a baseline algorithm in the revised manuscript during the final revision stage.
>
> ---
> References:
>
> [1] Mehrabi, Ninareh, et al. "A survey on bias and fairness in machine learning." _ACM Computing Surveys (CSUR)_ 54.6 (2021): 1-35.
>
> [2] Li, Tian, et al. "Fair resource allocation in federated learning." _arXiv preprint arXiv:1905.10497_ (2019).
>
> [3] Li, Tian, et al. "Federated optimization in heterogeneous networks." _Proceedings of Machine learning and systems_ 2 (2020): 429-450.
>
> [4] Wang, Yanmeng, Qingjiang Shi, and Tsung-Hui Chang. "Why Batch Normalization Damage Federated Learning on Non-IID Data?." _arXiv preprint arXiv:2301.02982_ (2023).
>
> [5] Chai, Zheng, et al. "Fedat: A communication-efficient federated learning method with asynchronous tiers under non-iid data." _ArXivorg_ (2020).
>
> [6] Zheng, Sihui, Cong Shen, and Xiang Chen. "Design and analysis of uplink and downlink communications for federated learning." _IEEE Journal on Selected Areas in Communications_ 39.7 (2020): 2150-2167.

---

### Decision · Action_Editors · 2023-08-23

**Recommendation:** Reject

**Comment:**

There is agreement that the overall topic of this work is interesting and relevant to the TMLR community. We encourage the authors to consider revising and resubmitting a new version of the work which more thoroughly explores the landscape of fairness vs. efficiency relative to competing approaches and performs ablations that help to explain where potential improvements of the approach arise. It would also be beneficial to continue iterating on the manuscript to carefully outline problem setting following the feedback from reviewers (e.g., precisely defining the notion of fairness considered, providing motivating examples showing that existing approaches suffer convergence degradation, etc) in order to make the motivation more clear from the beginning.

**Audience:**

The general topic of this work, which aims to explore efficiency and fairness in federated settings, would be of interest to the TMLR audience. As highlighted by the reviewers, the problem of convergence degradation resulting from fairness constraints is important for the practical implementation of federated learning systems, where efficiency is a key concern.

**Claims And Evidence:**

This work proposes AdaFedAdam, a method that seeks to improve fairness in federated settings without sacrificing convergence speed. All reviewers were in agreement that this is an interesting problem to consider.

However, there were shared concerns that the current results do not convincingly show such improvements. In particular, it would be beneficial to more comprehensively explore results across various hyperparameter settings (even just the method-specific ones) of competing approaches. Given that there are two axes the authors are considering---fairness vs. efficiency---it would be useful to show results for multiple potential hyperparameter configurations (resulting in trade-offs between fairness/convergence speed) in order to get as sense of the overall landscape of competing methods. For example, the authors could consider plotting the results of Figure 2 (# of rounds vs. accuracy and lower quantile accuracy) for several values of q in q-FFL, and several values of alpha in AdaFedAdam (potentially targeting different fairness regimes/quantiles---e.g., minimizing worst group accuracy, 10% worst group accuracy, 30% worst group accuracy). Currently we are only seeing a small snapshot of the possible fairness vs. efficiency landscape; getting a clearer picture of the trade-offs in multiple potential regimes is imperative to understand when AdaFedAdam would be useful relative to existing methods.

Additionally, all reviewers mentioned the need to perform more thorough ablation studies regarding the proposed approach to understand where benefits are coming from. For example, can you decouple the improvements from the objective vs. the optimization method in AdaFedAdam? What would happen if you solved the objective using another FedOpt variant? Alternatively, what would happen if you used FedAdam to solve other objectives such as the q-FFL objective? These are important research questions in order to understand the key contributions of the approach, and could also potentially help to extend the applicability of the work.

**Resubmission Of Major Revision:**

The authors may consider submitting a major revision at a later time.